# CCN1 interlinks integrin and hippo pathway to autoregulate tip cell activity

**Myo-Hyeon Park[1†], Ae kyung Kim[1,2†], Sarala Manandhar[1], Su-Young Oh[1], Gun-Hyuk Jang[1,2], Li Kang[1], Dong-Won Lee[3], Do Young Hyeon[4], Sun-Hee Lee[1], Hye Eun Lee[1], Tae-Lin Huh[2], Sang Heon Suh[5], Daehee Hwang[6], Kyunghee Byun[7], Hae-Chul Park[3], You Mie Lee[1,2]\***

[1]BK21 Plus KNU Multi-Omics Creative Drug Research Team, Research Institute of Pharmaceutical Sciences, College of Pharmacy, Kyungpook National University, Daegu, Republic of Korea; [2]School of Life Sciences and Biotechnology, College of Natural Sciences, Kyungpook National University, Daegu, Republic of Korea; [3]Department of Biomedical Sciences, Korea University, Ansan Hospital, Ansan, Republic of Korea; [4]School of Interdisciplinary Bioscience and Bioengineering, POSTECH, Pohang, Republic of Korea; [5]Department of Internal Medicine, Chonnam National University Hospital, Gwangju, Korea; [6]Department of New Biology and Center for Plant Aging Research, DGIST, Daegu, Republic of Korea; [7]Gachon University, School of Medicine, Incheon, Republic of Korea

**\*For correspondence:**
lym@knu.ac.kr

[†]These authors contributed equally to this work

**Competing interests:** The authors declare that no competing interests exist.

**Abstract** CCN1 (CYR61) stimulates active angiogenesis in various tumours, although the mechanism is largely unknown. Here, we report that CCN1 is a key regulator of endothelial tip cell activity in angiogenesis. Microvessel networks and directional vascular cell migration patterns were deformed in *ccn1*-knockdown zebrafish embryos. CCN1 activated VEGFR2 and downstream MAPK/PI3K signalling pathways, YAP/TAZ, as well as Rho effector mDia1 to enhance tip cell activity and CCN1 itself. VEGFR2 interacted with integrin αvβ3 through CCN1. Integrin αvβ3 inhibitor repressed tip cell number and sprouting in postnatal retinas from endothelial cell-specific *Ccn1* transgenic mice, and allograft tumours in *Ccn1* transgenic mice showed hyperactive vascular sprouting. Cancer patients with high *CCN1* expression have poor survival outcomes and positive correlation with *ITGAV and ITGB3* and high *YAP/WWTR1*. Thus, our data underscore the positive feedback regulation of tip cells by CCN1 through integrin αvβ3/VEGFR2 and increased YAP/TAZ activity, suggesting a promising therapeutic intervention for pathological angiogenesis.
DOI: https://doi.org/10.7554/eLife.46012.001

## Introduction

Angiogenesis involves extensive remodelling of the extracellular matrix (ECM) and endothelial cells (ECs) (*Stupack and Cheresh, 2002*). The ECM is important for dynamic and multifunctional regulation of cell behaviours and can modulate the bioavailability and activity of growth factors, cytokines, and extracellular enzymes (*Aszódi et al., 2006*). In addition, ECM proteins directly interact with cell surface receptors to activate signal transduction (*Bornstein and Sage, 2002*). Matricellular proteins are ECM proteins that do not contribute directly to the formation of these structures in vertebrates but are involved in cell–matrix interactions and other diverse cellular functions (*Bornstein, 1995*). Such proteins include thrombospondin-1, osteonectin, and members of the CYR61/CTGF/NOV (CCN) family of proteins (*Lau and Lam, 1999*). Within the CCN protein family, cysteine-rich angiogenic inducer 61 (CYR61, also known as CCN1) is a secreted protein that is a target of the YAP/TAZ Hippo signalling molecules (*Zhang et al., 2011*; *Yu et al., 2012*). CCN1 has been reported to

mediate a variety of cellular processes, including adhesion, chemotaxis stimulation, survival, and angiogenesis, in a cell type-dependent manner (*Grzeszkiewicz et al., 2002*; *Lin et al., 2004*). CCN1 also regulates the expression and activities of angiogenic factors, such as vascular endothelial growth factor (VEGF), in fibroblasts and osteoblasts (*Chen et al., 2001*; *Ivkovic et al., 2003*; *Athanasopoulos et al., 2007*; *Dean et al., 2007*). It is required for re-endothelialisation and wound repair in contexts where angiogenesis is indispensable (*Athanasopoulos et al., 2007*; *Bär et al., 2010*). The angiogenic behaviour of CCN1 has been attributed to its binding to integrin αvβ3, a major integrin expressed in ECs (*Leu et al., 2002*). *Ccn1*-null mice suffer embryonic lethality due to insufficient placental and embryonic blood vessel integrity (*Babic et al., 1998*), and *Ccn1*-knockdown in mouse embryos causes atrioventricular valve and septal defects due to apoptosis in the cushion tissue and reduction in gelatinase activity (*Mo and Lau, 2006*); this phenotype overlaps with VEGF-knockout phenotypes in the cardiovascular system (*Dor et al., 2001*). However, CCN1 is not expressed much in ECs but is rather expressed in other parenchymal or mesenchymal cells (*Su et al., 2004*), and thus, an EC-specific knockout strategy is likely not suitable for identification of the role of CCN1 in sprouting angiogenesis.

YAP/TAZ are overexpressed in various cancer tissues and have been identified as prognostic markers (*Wang et al., 2013*; *Han et al., 2014*). Immunohistochemistry (IHC) of tumours has revealed prominent expression and nuclear localization of YAP/TAZ in correlation with malignant features and poor patient outcomes (*Zanconato et al., 2016*). *CCN1* is a well-known transcriptional target of YAP/TAZ (*Zhao et al., 2008*), is secreted in cancer cells, and induces extensive tumour angiogenesis, underscoring its paracrine angiogenic effect (*Harris et al., 2012*; *Maity et al., 2014*). Thus, targeting abnormally activated YAP/TAZ is a promising strategy for the suppression of tumour progression, metastasis, and cancer relapse (*Corvaisier et al., 2016*; *Warren et al., 2018*). In addition, activation of YAP/TAZ by VEGF, a known angiogenic factor, facilitates expression of CCN1 (*Wang et al., 2017*). The presence of YAP in embryonic retinal vessels, along with reduced retinal vascular sprouting and decreased numbers of vascular branches upon EC-specific deletion of embryonic YAP/TAZ, has further emphasised the importance of YAP/TAZ in vascular development (*Choi and Kwon, 2015*; *Sakabe et al., 2017*).

Three types of functionally different ECs participate in the angiogenic process: tip cells, stalk cells, and phalanx cells (*Eilken and Adams, 2010*). All of these are engaged in the processes of vascular maturation and the maintenance of vascular integrity, thereby optimising blood flow, tissue perfusion, and oxygenation (*Eilken and Adams, 2010*). Tip cells are characterised by their position at the very tops of angiogenic sprouts and have extensive filopodial protrusions directed toward angiogenic attractants. Tip cells have a specific molecular signature, characterised by the expression of vascular endothelial growth factor receptor 2 (VEGFR2), VEGFR3, and DLL4. It has been reported that the VEGF gradient is important in the selection and induction of endothelial tip cells. Binding of VEGFR2 induces a signalling cascade that enables the activation of Notch-Delta signalling via DLL4 expression in ECs, converting them into tip cells; however, mechanism of sustained tip cell activity other than VEGF-mediated signalling has not yet been elucidated.

Here, we report that CCN1 plays crucial role as an auto-inducer of tip cell fate that stimulate angiogenesis through the interplay of YAP/TAZ signalling with the integration of integrin αvβ3-VEGFR2, suggesting a promising approach for the treatment of pathological angiogenesis facilitated by extensive stimulation of tip cells.

## Results

### CCN1 promotes sprouting angiogenesis in zebrafish

Secreted CCN1 is reported to facilitate EC migration and tumour angiogenesis via a paracrine effect (*Harris et al., 2012*; *Maity et al., 2014*), and YAP, an upstream regulator of CCN1, is expressed in the developing front of mouse retinal vessels (*Chintala et al., 2015*). Thus, to examine the precise mechanistic involvement of CCN1 in vascular formation, we designed two kinds of morpholino (MO) to target the transcription start site (ATG MO) or intron 1/exon 2 boundary of the *ccn1* gene (Splicing MO) (*Figure 1A*) and observed vascular development in TG (*flk1:EGFP*) embryos. MO-mediated knockdown of *ccn1* caused the formation of small heads, oedema, and bent trunk regions (*Figure 1B*). In TG (*flk1:EGFP*) embryos, control MO injection resulted in the formation of T-shaped

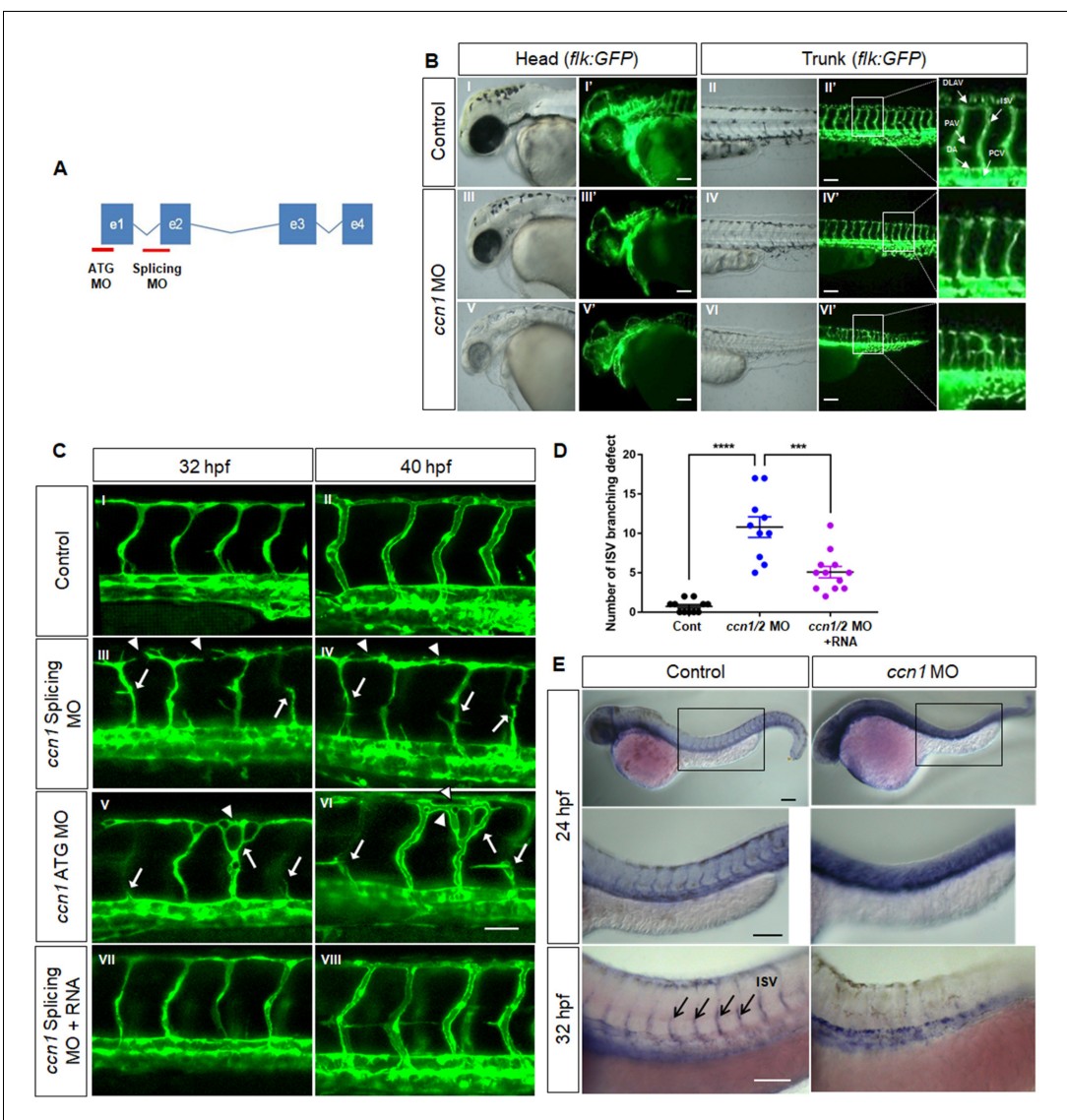

**Figure 1.** CCN1 knockdown induces defects in vessel migration and sprouting as well as identity of endothelial cells in zebrafish vascular development. (**A**) Scheme showing *ccn1* gene exon and intron. Morpholinos (MO) binding to *ccn1* ATG starting site (ATG MO) and intron 1/exon two boundary region (Splicing MO) denoted as red lines were synthesised and injected into one-cell-stage zebrafish embryos. (**B**) Lateral view of Tg(*Flk;EGFP*) embryos in head and trunk regions of control (I–II for bright field, I'–II' for dark field) and *ccn1* morphants (MO) (III–VI for bright field, III'–VI' for dark field) at 40 hpf. Arrows indicates sprouting vessels. Arrows indicate dorsolateral anastomose vessels (DLAV), intersegmental vessels (ISV), parachordal vessel (PAV), dorsal aorta (DA) and pericardial vein (PCV). The white square was enlarged and placed next to it. Scale bar = 25 μm (**C**) Anterior regions in trunk vessels were magnified at 32 and 40 hpf, in control, *ccn1* MO-treated (III-VI) and *ccn1* RNA rescued (VII-VIII) embryos. Arrows indicate misdirected or disconnected vessels and arrowheads do disconnected or abnormal DLAV. Scale bar = 50 μm. (**D**) Number of intersomitic vessel (ISV) defect were counted in each embryo and graphed. (**F**) Whole mount in situ hybridization was performed with control and *ccn1* MO embryos at 24 hpf and 32 hpf with *zFlk1* antisense probe. Purple colour shows expression of *zFlk1* mRNA expression. Arrows indicates ISVs. Scale bar = 100 μm.

DOI: https://doi.org/10.7554/eLife.46012.002

The following figure supplement is available for figure 1:

**Figure supplement 1.** Arterio-venous identity was altered in *ccn1* MO embryos.

DOI: https://doi.org/10.7554/eLife.46012.003

intersegmental vessels (ISVs) and dorsolateral anastomose vessels (DLAVs) at 40 hr post-fertilisation (hpf) (I′ and II′ in *Figure 1B*), whereas the vasculature in the head and trunk regions of *ccn1* morphants was abnormally sprouted and disconnected (III′, IV′, V′, and VI′ in *Figure 1B*). *ccn1* morphants lost the T-shaped morphology previously displayed at the DLAV and ISV connexions (IV′ and VI′ in *Figure 1B*). When we observed more precisely in ISVs, in control animals, frontal cells from the DA migrated along a left-ascending path to the parachordal vessel (PAV) and then along a right-ascending path to the DLAV (I and II, arrows in *Figure 1B*); conversely, in two kinds of *ccn1* morphants, these cells took a right and then left-ascending or bifurcating path to DLAV or not migrating from DAV (III-VI, arrows in *Figure 1C*, *Figure 1D*), and disconnected or malformed DLAV (III-VI arrowheads in *Figure 1C*, *Figure 1D*) were significantly increased at both 32 and 40 hpf. However, injection of sense RNA of *ccn1* significantly rescued the vascular malformations and altered phenotypes induced by *ccn1* morpholinos (*Figure 1C* VII and VIII), suggesting that CCN1 is an essential factor for the vascular development in zebrafish. In addition, *ccn1* morphants demonstrated ectopic expression of *flk-1*, which was expressed not in vascular ECs but across the entire anterior–posterior body axis (*Figure 1E*). Aortic vessels disappeared and changed into venous types as detected by in situ expression of *Notch-1b* (aortic marker) and *Flt-4* (venous marker) (*Figure 1—figure supplement 1*). Therefore, the results suggest that CCN1 is essential for the normal migration and sprouting as well as the identity of ECs in zebrafish vascular development.

## CCN1 promotes endothelial sprouting activity in angiogenesis

CCN1 is expressed during active angiogenic stages in postnatal retina tip cells, and its expression decreases upon EC maturation (*Chintala et al., 2015*). However, the above experiment raised the question of why CCN1 deletion increases sprouting and whether CCN1 can stimulate or inhibit tip cell activity. We used recombinant CCN1 protein to mimic the secreted protein ex vivo and performed a mouse aortic ring assay, as well as an EC spheroid sprouting assay in vitro, using VEGF-A (VEGF) as a positive control. CCN1 induced intensive vascular sprouts from aortic explants and significantly increased tip cell populations to the same degree as VEGF, as determined by tip cell markers, DLL4 and CD34 (*Siemerink et al., 2012*) as well as an EC marker, CD31 (*Figure 2A*). CCN1 also significantly increased sprouting branches from spheroids, as determined by both lengths and numbers of tip cells after immunostained with DLL4 and CD34 antibodies (*Figure 2B*). Filopodial formation, which is an important marker of tip cells in migrating cells, was also detected in vitro by immunostaining with phalloidin. In human ECs, filopodia markedly increased upon exposure to CCN1 (*Figure 2C*). To investigate the angiogenic activity of CCN1, we performed in vitro angiogenesis, tube formation, and EC migration assays. CCN1 markedly increased capillary tube formation (*Figure 2D*) and promoted EC migration in the wound migration assay (*Figure 2E*, *Videos 1* and *2*). Because tip and stalk cell activities are increased during active angiogenesis (*Eilken and Adams, 2010*), we checked whether the expression of marker genes for tip and stalk cells changed following CCN1 treatment in human umbilical vein ECs (HUVECs). Expression levels of tip cell marker genes like *VEGFR2*, *DLL4* (*Hellström et al., 2007*), and *SOX17* (*Lee et al., 2014*) were elevated, but those of stalk cell marker genes like *JAG1*, *ROBO4*, and *NOTCH1* were not changed by CCN1, nor by VEGF (*Figure 2F*). Protein levels of VEGFR2, DLL4, and SOX17 were also increased by CCN1 (*Figure 2G*). To confirm these results, *DLL4* promoter-driven luciferase (*DLL4-Luc*) reporter activity was observed. *DLL4-Luc* reporter activity was significantly increased by CCN1, as by VEGF (*Figure 2H*). These results clearly suggest that CCN1 induces tip cell activity and identity in sprouting during angiogenesis.

## CCN1-induced YAP/TAZ activation via VEGFR2 signalling is critical for tip cell activity

YAP/TAZ have critical roles in vascular sprouting (*Kim et al., 2017*; *Park and Kwon, 2018*), and their activation promotes tip cell activity and expression of target genes, such as *Ccn1* (*Sakabe et al., 2017*). Thus, the probable involvement of VEGFR2 signalling in the regulation of YAP/TAZ was investigated in CCN1-treated ECs. YAP and TAZ translocation to the nucleus was significantly upregulated by treatment with CCN1 for 1 hr in a dose-dependent manner (*Figure 3A*). Phosphorylation of YAP/TAZ at S127/89, respectively, leads to cytosolic degradation, thus inhibiting their nuclear localization. In growth factor-mediated YAP signalling via the MAPK and PI3K nexus, large tumour

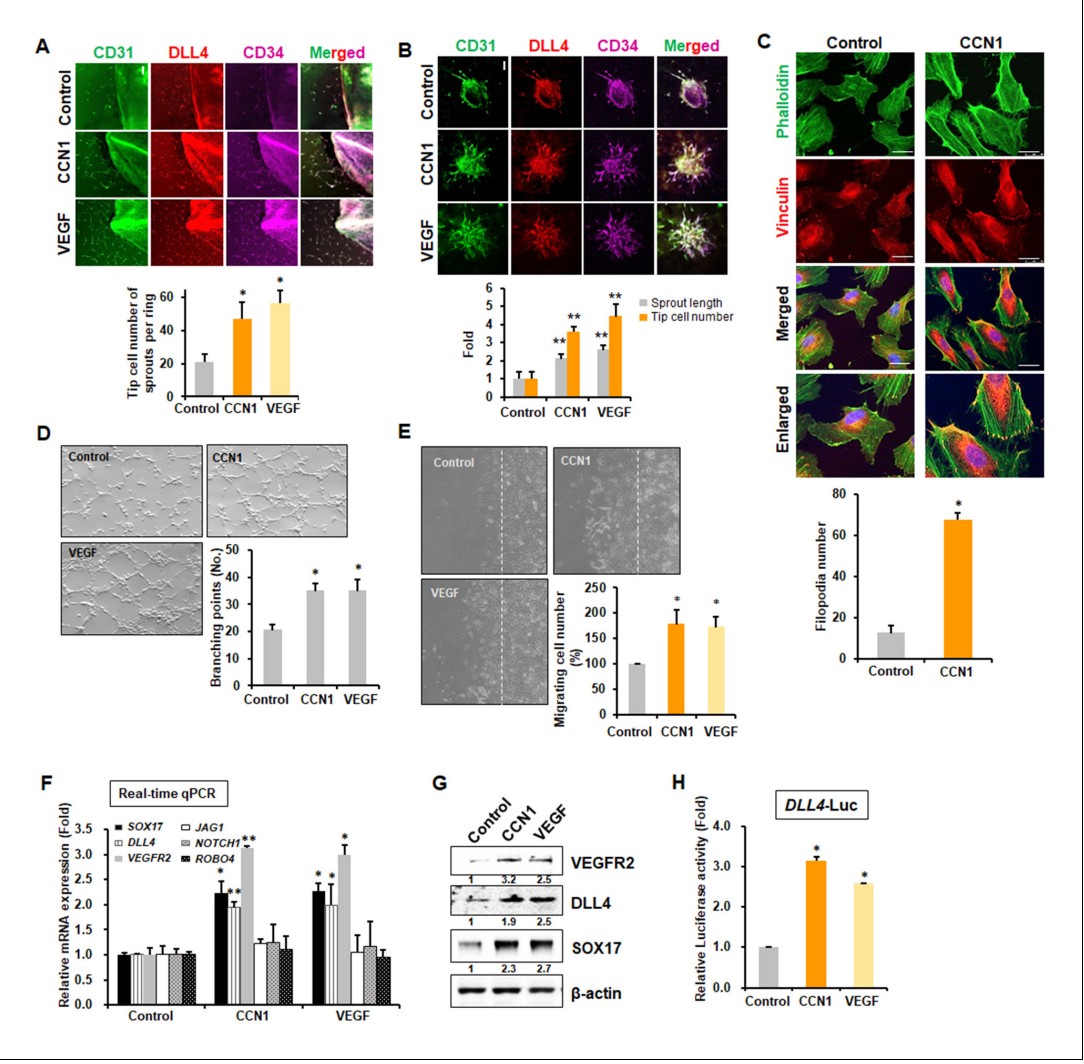

**Figure 2.** CCN1 stimulates tip cell activity in ECs. (**A**) Mouse aorta isolated and cut at 6 weeks old, planted on Matrigel-coated plate and incubated for 7 days in presence or absence of CCN1 (10 ng/mL) or VEGF (10 ng/mL). Expression of CD31, DLL4, and CD34 were visualised with respective antibodies by the immunofluorescent assay under the confocal microscope. Tip cells expressing CD31+/DLL4+/CD34+ cells were counted and graphed. Three independent experiments were performed (n = 3/group). *p<0.05 vs control. Scale bar = 100 µm. (**B**) HUVEC spheroids were cultured in presence or absence of CCN1 (10 ng/mL) or VEGF (10 ng/mL) for 48 hr. Expression of CD31, DLL4, and CD34 were visualised with respective antibodies by the immunofluorescent assay under the confocal microscope. Length of sprout and tip cells expressing CD31+/DLL4+/CD34+ cells were counted and graphed (n = 3/group). **p<0.01 vs Control. Scale bar = 100 µm. (**C**) Filopodia number per field was counted in HUVEC cells treated with PBS (Control) or CCN1 (10 ng/mL) for 30 min and imaged using confocal microscope after staining with vinculin (red) and phalloidin (green). *p<0.0001 vs. Control. Scale bar = 25 µm. (**D**) Branching points of tubular networks were counted in HUVECs cultured in the presence or absence of CCN1 or VEGF for 48 hr (n = 3/group). Magnification = 40 × . *p<0.05 vs. control. (**E**) Confluent HUVECs were wounded and cultured in the presence or absence of CCN1 or VEGF for 6 hr. Migrating cells from wound line (dotted line) were counted (n = 3/group). Magnification = 40 × . *p<0.05 vs. control. (**F**) HUVECs were treated with CCN1 (10 ng/mL) or VEGF (10 ng/mL) for 24 hr, and total RNA was collected for real-time qRT-PCR analysis. Expression of tip cell marker genes (*SOX17, DLL4, VEGFR2*) and stalk cell marker genes (*JAG1, NOTCH1, ROBO4*) was analyzed. *p<0.05 or **p<0.01 vs. Control. (**G**) Western blotting analysis of VEGFR2, DLL4, and SOX17 protein levels in the same samples as used in **F**. β-actin was used as an internal control. Fold increase was indicated as a number below each band. (**H**) Luciferase assays in HUVECs transfected with *DLL4-Luc* reporter vector and treated with CCN1 or VEGF (10 ng/mL) for 24 hr. Luciferase activity was normalised to *Renilla* luminescence. *p<0.001 vs. Control.

*Figure 2 continued on next page*

*Figure 2 continued*

DOI: https://doi.org/10.7554/eLife.46012.004

The following source data is available for figure 2:

**Source data 1.** Source data for *Figure 2G*.

DOI: https://doi.org/10.7554/eLife.46012.005

suppressor kinase 1 (LATS1), a tumour suppressor that phosphorylates YAP/TAZ, is repressed through reduced phosphorylation, thus enhancing the nuclear localization of YAP (*Gumbiner and Kim, 2014*; *Kim and Gumbiner, 2015*). We found that levels of phosphorylated LATS were decreased maximally at 1 hr and those of phosphorylated YAP/TAZ were decreased from 15 min, maximally at 1 hr by CCN1 in HUVECs (*Figure 3B*), indicating that CCN1 activates YAP/TAZ in ECs at the early time after stimulation.

VEGFR2 signalling is reportedly linked to YAP/TAZ activation in developmental angiogenesis (*Wang et al., 2017*). VEGFR2 activation is a key event in determining the fate of an EC as a tip cell (*Carmeliet et al., 2009*). Thus, we investigated whether VEGFR2 is activated by CCN1 in ECs. CCN1 induced the phosphorylation of VEGFR2 from 10 min at slightly earlier time point of YAP/TAZ activation within 1 hr (*Figure 3C*) and in a dose-dependent manner (*Figure 3D*). VEGFR2 activation after serum stimulation (*Figure 3—figure supplement 1A*) (*Latinkic et al., 2001*) was highly repressed upon siRNA-mediated *CCN1* knockdown (siCCN1) (*Figure 3E*), suggesting that CCN1 activates VEGFR2. To confirm whether CCN1 is able to activate VEGFR2 in the absence of endogenous VEGF, we transfected HUVECs with siVEGF and found that CCN1 activated VEGFR2 regardless of endogenous VEGF (*Figure 3—figure supplement 1B*). Knockdown of *CCN1* with siCCN1 repressed the mRNA expression of the tip cells markers, *DLL4* and *VEGFR2* (*Figure 2* and *Figure 3—figure supplement 1C*) as well as the migration and sprouting ability (*Figure 3—figure supplement 1D*), indicating that endothelial CCN1 is important for EC tip cell activity. The phosphorylation of Y1054/1059 in the VEGFR2 kinase domain mediates kinase activity, and that of Y1175 mediates binding to the p85 subunit of PI3K and to MAPKs, which is involved in migration as well as proliferation (*Olsson et al., 2006*). Recently, it has been reported that VEGFR activation induces downstream kinases to inhibit LATS function and activate YAP/TAZ (*Azad et al., 2018*). Therefore, we checked that extracellular signal-regulated kinase (ERK), p38 MAPK, and p85 PI3K, all downstream targets of Y1175 phosphorylation, were activated by CCN1 (*Figure 3F*). Next, to confirm whether CCN1-mediated tip cell activity involves the MAPK and PI3K signalling cascades, we used pharmacological inhibitors for ERK1/2 (PD98059), p38 MAPK (SB203580), JNK MAPK (SP600125), and PI3K (LY294002). Upon treatment with all inhibitors except SP600125, *DLL4* mRNA and protein levels were highly repressed (*Figure 3G and H*). To confirm these results, filopodial formation, as determined by phalloidin staining, was highly reduced in ECs exposed to CCN1 together with the inhibitors (*Figure 3I*).

CCN1-mediated increases in the phosphorylation of STAT-1 and STAT-3 further implicate its probable involvement in VEGFR2 activation, similar to that of VEGF (*Bartoli et al., 2000*; *Bartoli et al., 2003*) (*Figure 3—figure supplement 1E*). Thus, collectively, these results suggest that CCN1 induces tip cell activation through the VEGFR2/ERK1/2 and p38MAPK/PI3K signalling pathways in ECs.

Next, we determined whether the CCN1-induced MAPK and PI3K signalling pathways are involved in the activation of YAP/TAZ. The phosphorylation of LATS1 and YAP/TAZ was decreased by CCN1, but the PD98059, SB203580, and LY294002 inhibitors restored phosphorylation to control levels, thus confirming the repression of phosphorylated LATS1 in ECs, resulting in YAP/TAZ activation by CCN1

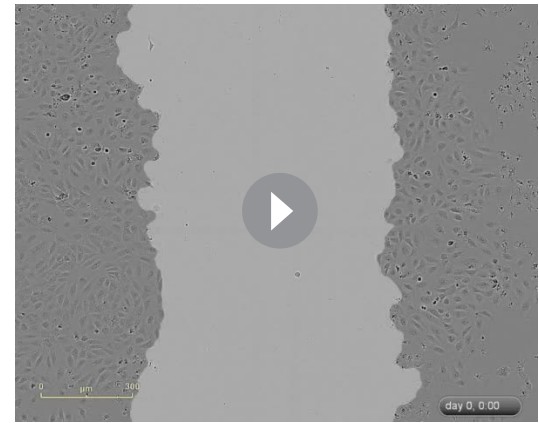

**Video 1.** CCN1 promoted EC migration.

DOI: https://doi.org/10.7554/eLife.46012.006

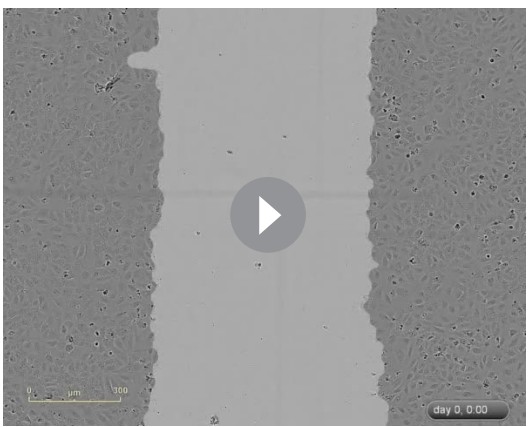

**Video 2.** CCN1 promoted EC migration.
DOI: https://doi.org/10.7554/eLife.46012.007

(*Figure 3J*). In support of these findings, expression of YAP/TAZ target gene *CCN1* was also significantly increased by CCN1 treatment and repressed by PD98059, SB203580, and LY294002 (*Figure 3K*), suggesting the involvement of MAPK and PI3K as well as the expression of CCN1 via the activation of YAP/TAZ in ECs. An inhibitor of YAP/TAZ, verteporfin (VP), inhibited VEGFR2, CCN1, and DLL4 expression induced by CCN1, confirming the involvement of YAP/TAZ in tip cell activity (*Figure 3L*). These results suggest that CCN1 activates the VEGFR2-MAPK/PI3K-YAP/TAZ pathway to facilitate the turnover of CCN1, further sustaining tip cell activity in ECs.

## mDia1 is involved in CCN1-induced filopodial formation and YAP/TAZ activation in ECs

Polymerization of actin and the fast-growing end of actin filaments is a crucial step in filopodial formation, and mDia1, a formin family protein, is essential for Rho GTPase-induced filopodial formation (*Evangelista et al., 2003*; *Fan and Mellor, 2012*). Therefore, we investigated whether mDia1 is involved in CCN1-induced filopodial formation in ECs. We found that CCN1 increased *DIAPH1* mRNA expression by 1.5-fold (*Figure 4A*) and also overexpression of *DIAPH1 (mDia1)* increased filopodial numbers similar to that in CCN1-treated HUVECs (*Figure 4B*). However, *DIAPH1* dominant-negative mutation decreased CCN1-induced filopodial formation (*Figure 4B*), suggesting that mDia1 is essential for CCN1-induced filopodial formation in ECs. Since the small Rho GTPase Cdc42 is critical for EC motility by promoting endothelial focal adhesions and filopodial formation by regulating actomyosin contractility (*Barry et al., 2015*) and tip cell migration (*Sakabe et al., 2017*), we assessed levels of activated Cdc42 GTPases. The active form of Cdc42 assayed by pull-down and immunoprecipitation (IP) analysis as described previously (*Wang et al., 2003*) was increased by CCN1 treatment and *DIAPH1* overexpression but not the *DIAPH1* mutant with CCN1 (*Figure 4C*). Phosphorylated N-Wasp, induced by active Cdc42 (*Sakabe et al., 2017*), was also detected in CCN1-treated and *DIAPH1*-overexpressing cells but not in *DIAPH1* mutant-expressing cells (*Figure 4D*), suggesting that mDia1 is required for CCN1-induced Cdc42 activation in tip cell formation. Next, we further identified whether CCN1-induced mDia1 activation is responsible for the YAP/TAZ activation. Nuclear localization of YAP/TAZ was induced by overexpression of *DIAPH1* as well as CCN1 treatment, but not by the *DIAPH1* DN mutants (*Figure 4E*), indicating that mDia1 is necessary for YAP/TAZ nuclear translocation in CCN1-induced tip cell formation.

## Integrin αvβ3 is a key mediator of hippo pathway induced by CCN1

Integrins involved in various functions, are induced by CCN1, and are also reported to activate VEGFR2 and its downstream signalling pathway (*Chen and Lau, 2009*). Integrins αvβ3 and αvβ1 are involved in EC migration induced by CCN1 (*Chen and Lau, 2009*) and are the major vitronectin receptors on ECs. As the ability of ECs to adhere to vitronectin in the presence of CCN1 was completely blocked by the addition of β3 antibody at 45 min (*Figure 5—figure supplement 1A*), αvβ3 may be the major receptor mediating adhesion of CCN1 to ECs (*Kireeva et al., 1998*). We also found that integrin β1 was not involved in the adhesion of fibronectin to ECs (*Figure 5—figure supplement 1A*). To confirm whether integrin αvβ3 is essential for the induction of tip cells by CCN1, we first checked whether CCN1 regulates integrin β3 expression. CCN1, as well as VEGF increased β3 integrin expression according to western blotting (WB) and immunofluorescence (IF) assays (*Figure 5—figure supplement 1B,C*), suggesting autocrine regulation of CCN1 angiogenic signalling through integrin αvβ3.

Next, to gain insight into the molecular involvement of integrin αvβ3 in CCN1-induced tip cell activity, we investigated whether integrin αvβ3 interacts with VEGFR2 via CCN1. IP and WB analyses

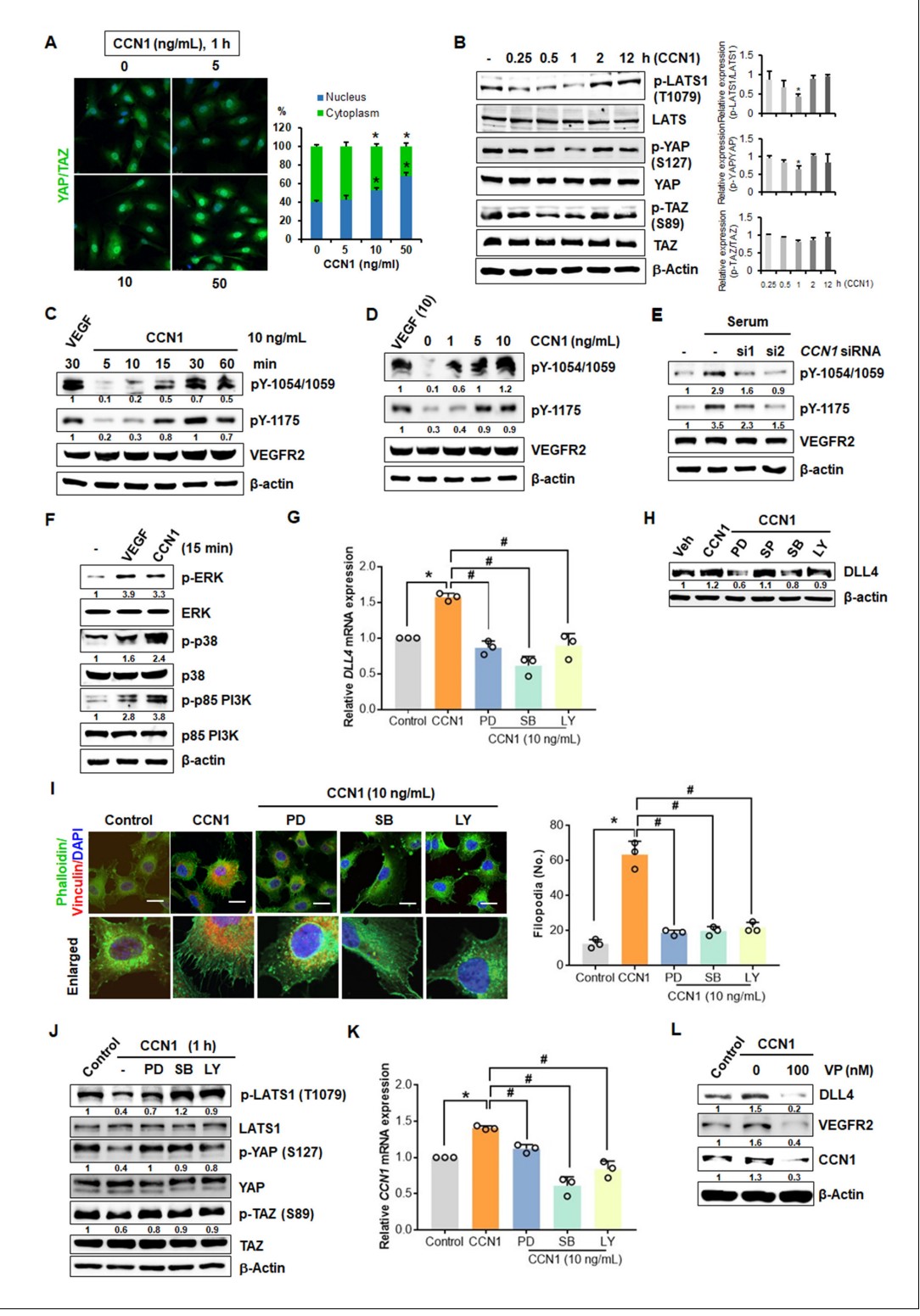

**Figure 3.** CCN1-VEGFR2-MAPK, PI3K-YAP/TAZ signalling involved in CCN1-mediated tip cell activation. (**A**) Nuclear localization of YAP/TAZ in HUVECs treated with CCN1 at 5, 10, or 50 ng/mL for 1 hr. (**B**) Western blotting of LATS, YAP, and TAZ and their phosphorylated forms in HUVECs treated with CCN1 (10 ng/mL) for indicated times. Quantitation for phosphorylated protein level was normalised by each protein and graphed shown in right panel. *p<0.05 vs. Control. (**C**) Western blotting of phosphorylated VEGFR2 at tyrosine (Y)−1054/1059 and Y1175 in HUVECs treated with VEGF (10 ng/mL) or CCN1 (10 ng/mL) for 5, 10, 15, 30, or 60 min. β–actin was used as a

*Figure 3 continued on next page*

*Figure 3 continued*

loading control. (**D**) Western blotting of VEGFR2 phosphorylated at Y1054/1059 and Y1175 in HUVECs treated with VEGF (10 ng/mL) or CCN1 (0, 1, 5, or 10 ng/mL) for 30 min. β–actin was used as a loading control. (**E**) Western blotting of VEGFR2 phosphorylated at Y1054/1059 and Y1175 in HUVECs transfected with *CCN1* siRNA. β–actin was used as a loading control. (**F**) Western blotting of phosphorylated or natural forms of p-ERK, p38 MAPK, and p85 PI3K in HUVECs were treated with VEGF or CCN1 (10 ng/mL) for 15 min. (**G, H**) *DLL4* mRNA (**G**) and protein (**H**) levels in EA.hy926 cells treated with PBS (Control) or CCN1 (10 ng/mL) alone or together with MAPK kinase inhibitors PD98059, SP600125, or SB203580 (20 mM) or PI3K inhibitor LY294002 (20 mM) for 24 hr. *p<0.005 vs. Control, #p<0.0001 vs. CCN1. (**I**) Filopodia number per field was quantified after staining with anti-vinculin antibody (red) and phalloidin (green) in EA.hy926 cells treated with PBS (Control) or CCN1 (10 ng/mL) alone or together with PD98059 (20 mM), SP600125 (20 mM), SB203580 (20 mM), or LY294002 (20 mM) for 30 min. Scale bar = 10 μm. *p<0.0001 vs. Control, #p<0.0001 vs. CCN1. (**J**) Western blotting of indicated proteins in EA.hy926 cells treated with PBS (Control) or CCN1 (10 ng/mL) alone or together with PD98059 (20 mM), SB203580 (20 mM), or LY294002 (20 mM) for 1 hr. (**K**) *CCN1* mRNA expression determined by qRT-PCR of total RNA obtained from the same groups as in **G** but treated with CCN1 for 24 hr. *p<0.005 vs. Control, #p<0.005 vs. CCN1. (**L**) Detection of DLL4, VEGFR, and CCN1 proteins in HUVECs treated with CCN1 (10 ng/mL) for 24 hr in the presence or absence of verteporfin (VP, 100 nM) for 24 hr. Fold changes were noted under each protein band in C-F, H, J and L.

DOI: https://doi.org/10.7554/eLife.46012.008

The following source data and figure supplement are available for figure 3:

**Source data 1.** Source data for *Figure 3B, C, D, E, F, H, J, L*.
DOI: https://doi.org/10.7554/eLife.46012.010

**Figure supplement 1.** Knockdown of *CCN1* decreased tip cell markers.
DOI: https://doi.org/10.7554/eLife.46012.009

---

showed strong interactions between VEGFR2 and integrin αvβ3, as well as their interaction with CCN1 upon CCN1 treatment for 24 hr. However, cell exposure to the αvβ3 inhibitor peptide cyclo (RGDfK) blocked the interactions of VEGFR2 with integrin αvβ3, CCN1 with integrin αvβ3, and CCN1 with VEGFR2 (*Figure 5A*). In addition, in situ proximity ligation assay (PLA) also clearly revealed increased interactions among CCN1/integrin αvβ3/VEGFR2 in ECs compared to that in controls (*Figure 5B* and *Figure 5—figure supplement 1D*). CCN1-induced VEGFR2 downstream kinase signalling activity was inhibited by cyclo(RGDfK) (*Figure 5C*). We also determined whether integrin αvβ3 is essential for YAP/TAZ activation by CCN1. Co-treatment with cyclo(RGDfK) rescued the decreased phosphorylation of YAP/TAZ by CCN1 (*Figure 5D*). At the same time, increased YAP/TAZ nuclear localization in the CCN1-treated group was blocked by cyclo(RGDfK) co-treatment (*Figure 5E*), followed by repressed *DLL4-Luc* activity and expression of tip cell marker genes *DLL4* and *SOX17* (*Figure 5F and G*). These results suggest that CCN1-mediated tip cell activation is regulated by YAP/TAZ through the integrin αvβ3/VEGFR2 signalling pathway.

We then repeated the aortic ring and EC spheroid sprouting assays in the presence of cyclo (RGDfK). In aortas treated with cyclo(RGDfK), CCN1-induced tip cell numbers were significantly decreased CD31+/DLL4+/CD34+ tip cells in *Figure 5H*. CCN1 increased the sprouting length and tip cell number by 2- and 3.5-fold, respectively, when compared to control group, and this was abolished by cyclo(RGDfK) in cultured spheroids (*Figure 5I*).

## Endothelial CCN1 via integrin αvβ3 is crucial for vascular sprouting and growth in vivo.

Because *Ccn1* mRNA expression is much lower in ECs than in other mesenchymal cells or some epithelial cells (*Su et al., 2004*) but *Ccn1* mRNA is expressed in the front region of developing retinal vessels, to investigate the role of CCN1 in ECs in vivo, we generated EC-specific TG mice rather than knockout mice. Visualisation of developing vessels by IF staining with isolectin B4 and CCN1 antibodies revealed that CCN1 was not highly expressed in retinal ECs at postnatal day 6 (P6) (*Figure 6—figure supplement 1*) (*Chintala et al., 2015*). In EC-specific *Ccn1*-TG mice, we confirmed that CCN1 was expressed in retinal ECs (*Figure 6—figure supplement 1*), together with significant increases in retinal vessel branching and number of tip cells with filopodia (*Figure 6A,B*). However, in each retinal quadrant, the radial outgrowth of retinal vessels from the optic nerve head to the peripheral vascular front was similar to that observed in wild-type (WT) mice (*Figure 6A*), indicating

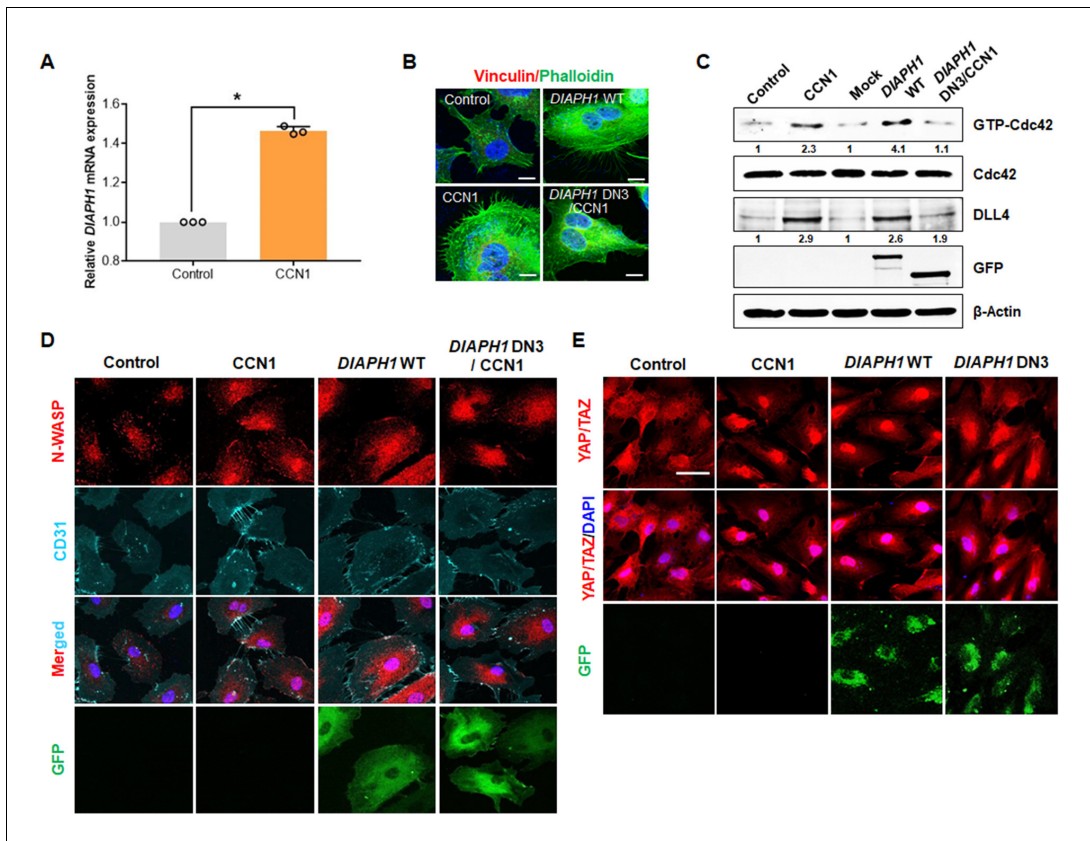

**Figure 4.** mDia1is essential for CCN1-induced Cdc42 activation in tip cell formation. (**A**) HUVECs were treated with PBS (Control) or CCN1 (10 ng/mL) for 24 hr, and total RNA was used for the detection of *DIAPH1 (mDia1)* mRNA expression by qRT-PCR. *p<0.001 vs. Control. (**B**) After transfection with *DIAPH1* WT full length and *DIAPH1* double-negative mutant plasmids, HUVECs were treated with CCN1 and immunostained with anti-vinculin antibody (red) and phalloidin (green) to visualise filopodia. Scale bar = 10 μm. (**C, D**) HUVECs transfected with *DIAPH1* WT full length and *DIAPH1* double-negative mutant plasmids were treated with CCN1 (10 ng/mL) for 30 min, and active Cdc42 was assessed by western blotting analysis (**C**) and visualised by p-N-Wasp (**D**). Fold changes were noted under each protein band. (**E**) After HUVECs were transfected with *DIAPH1* WT full length and *DIAPH1* DN3 mutant plasmid, starved cells for 16 hr were treated with CCN1 10 ng/ml for 1 hr and detected YAP/TAZ by IF. Scale bar = 100 μm.

DOI: https://doi.org/10.7554/eLife.46012.011

The following source data is available for figure 4:

**Source data 1.** Source data for *Figure 4C*.
DOI: https://doi.org/10.7554/eLife.46012.012

a specific role for CCN1 in endothelial branching and tip cell formation in vivo. Following injection of cyclo(RGDfK) for 2 days, the number of sprouting tip cells with filopodia significantly decreased at P5 and P6 (*Figure 6B*). To further investigate the autoregulation of CCN1 in sprouting angiogenesis in vivo, we performed aortic ring and Matrigel invasion assays in *Ccn1*-TG mice. The aortic rings isolated from these TG mice showed an increased number of tip cells and sprouting ECs compared to those in WT mice, and these were significantly reduced in cyclo(RGDfK)-treated aortas (*Figure 6C*). In addition, cyclo(RGDfK) treatment repressed in vivo angiogenesis in the Matrigel plug assay. Haematoxylin and eosin (H and E) staining of sections of plugs from *Ccn1*-TG mice revealed an increased number of neovessels and microvessels compared to that in controls, while in Matrigel plugs treated with cyclo(RGDfK), both neovessel and microvessel densities were decreased compared to those in *Ccn1*-TG mice without cyclo(RGDfK) (*Figure 6D*). Similar results were observed in the CAM assay, with results of treatment with CCN1 alone or together with cyclo(RGDfK) further strengthening the above finding (*Figure 6—figure supplement 2A*). We also confirmed that treatment with SU5416, a

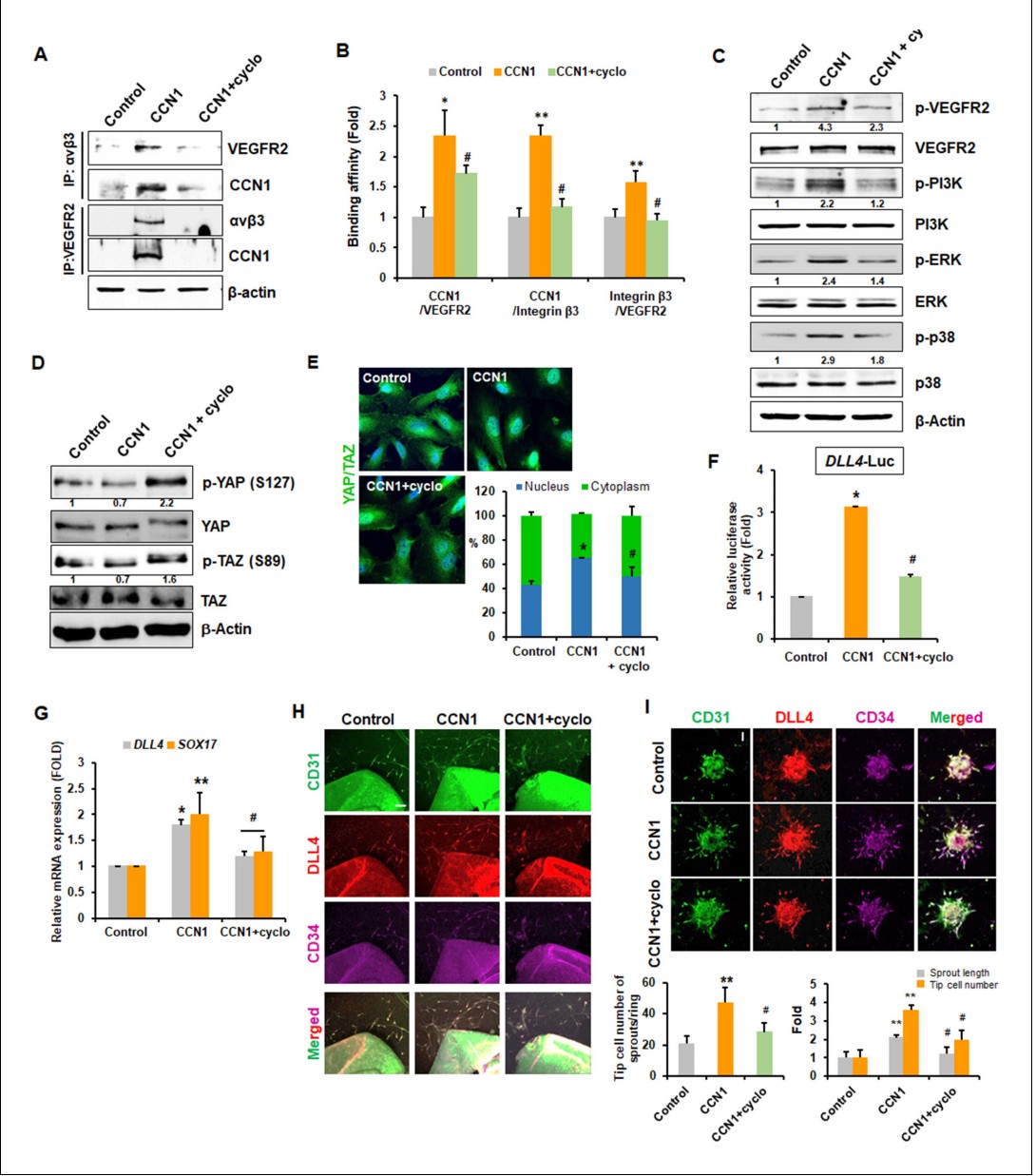

**Figure 5.** Tip cell properties are induced by CCN1 via interactions between integrin αvβ3 and VEGFR2. (**A**) HUVECs treated with CCN1 (10 ng/mL) for 24 hr in the presence or absence of cyclo(RGDfK) (1 μg/mL) were used for IP and immunoblotting analyses to detect interactions between integrin αvβ3 and VEGFR2 or Cy61. β-actin was used as an internal loading control. (**B**) Proximity ligation assay was performed, and double-fluorescent microscopic images showing protein–protein interactions in cells were quantified. *p<0.01, **p<0.05 vs. Control, #p<0.01 vs. CCN1. (**C, D**) HUVECs were treated with CCN1 (10 ng/mL) for 30 min (**C**) or 1 hr (**D**) in the presence or absence of cyclo(RGDfK) (1 μg/mL), and western blotting was performed for the indicated proteins. Fold changes were noted under each protein band. (**E**) The same samples as in D were analysed for the nuclear localization of YAP/TAZ via IF. *p<0.05 vs. Control, #p<0.05 vs. CCN1. (**F**) HUVECs were transfected with *DLL4-Luc* reporter vector and treated with CCN1 (10 ng/mL) for 24 hr in the presence or absence of cyclo(RGDfK) (1 μg/mL). Luciferase activity was normalised to *Renilla* luminescence. *p<0.01 vs. Control, #p<0.01 vs. CCN1. (**G**) Total RNA was obtained from HUVECs treated with CCN1 (10 ng/mL) for 24 hr in the presence or absence of cyclo(RGDfK) (1 μg/mL). Expression levels of *DLL4* and *SOX17* mRNA were determined by qRT-PCR. *p<0.01, **p<0.05 vs. Control, #p<0.01 vs. CCN1. (**H**) Aortic sprouting assays were performed and determined after incubation with CCN1 (10 ng/mL) in the presence or absence of cyclo(RGDfK) for 7 days. Expression of CD31, DLL4, and CD34 were visualised with their antibodies by the immunofluorescent assay under the confocal microscope. Tip cells expressing CD31+/DLL4+/CD34+ cells were counted and graphed. Three independent experiments were performed (n = 3/group).
*Figure 5 continued on next page*

*Figure 5 continued*

*p<0.05 vs control. Scale bar = 100 µm. (I) Spheroid assays were performed with CCN1 (10 ng/mL) in the presence or absence of cyclo(RGDfK) for 48 hr. Expression of CD31, DLL4, and CD34 were visualised with their antibodies by the immunofluorescent assay under the confocal microscope. Length of sprout and tip cells expressing CD31+/DLL4+/CD34+ cells were counted and graphed (n = 3/group). Scale bar = 100 µm **p<0.001 vs. Control, #p<0.05 vs. CCN1 only.
DOI: https://doi.org/10.7554/eLife.46012.013
The following source data and figure supplement are available for figure 5:
**Source data 1.** Source data for *Figure 5C and D*.
DOI: https://doi.org/10.7554/eLife.46012.015
**Figure supplement 1.** The involvement of integrin αvβ3 in CCN1-induced activation of EC.
DOI: https://doi.org/10.7554/eLife.46012.014

VEGFR2 kinase-specific inhibitor, demonstrated the same inhibitory results in CAM and Matrigel plug assays following treatment with CCN1 (*Figure 6—figure supplement 2A,B*). To rule out endogenous VEGF as a prime effector of angiogenesis in *Ccn1*-TG mice, we compared the plasma VEGF levels of the WT and TG mice. Levels of plasma VEGF did not differ between WT and EC-specific *Ccn1*-TG mice, and treatment of cultured ECs with CCN1 did not affect secreted VEGF levels (*Figure 6—figure supplement 2C,D*). Therefore, collectively, these results indicate that integrin αvβ3 and VEGFR2 are involved in CCN1-induced tip cell generation and sprouting during in vivo angiogenesis. Finally, we confirmed the expression patterns of VEGFR2 and integrin β3 in the mice. In *Ccn1*-TG mice, we observed significantly increased and co-localised expression of VEGFR2 and integrin β3 in the front region of the retinal vessels (*Figure 6E*). Taken together, these data indicate that CCN1 induces ECs to acquire tip cell activity via the activation of VEGFR2 signalling through integrin αvβ3.

## Role of CCN1 in pathological angiogenesis and clinical correlation with integrin/Hippo pathway

YAP/TAZ are overexpressed in various cancer tissues, and their nuclear localization has been identified as a potential prognostic marker (*Wang et al., 2013*; *Han et al., 2014*; *Zanconato et al., 2016*). To address the role of CCN1 in pathological angiogenesis, we used an allograft Lewis lung carcinoma (LLC) tumour model in EC-specific *Ccn1*-TG mice. Surprisingly, induced expression of *Ccn1* in ECs induced active angiogenic vessels, representing a significant decrease in pericyte coverage, both in the central and peripheral regions of the tumour (*Figure 7A*). *Ccn1* overexpression in ECs resulted in significant upregulation of VEGFR2 expression as well as increased sprouting activity in the central region of the tumour (*Figure 7B*). These results suggest that CCN1 induces active angiogenesis through VEGFR2 in pathological tumour vasculature.

Next, we investigated whether upregulated CCN1 expression is correlated with cancer patient survival and the expression of molecules involved in tip cell activity based on data from The Cancer Genome Atlas (TCGA). The overall survival rate was significantly higher in those with low expression of *CCN1* compared to that in the high expression group among lung squamous cell carcinoma (LUSC) (*Figure 7C*) as well as bladder urothelial cancer (BLCA) and stomach adenocarcinoma (STAD) patients (*Figure 7—figure supplement 1A,C*). In addition, we compared the expression of *ITGB3*, *VEGFR2*, *DLL4*, *YAP* and *WWTR1* in the *CCN1*-low and -high groups. *CCN1* expression was significantly positively correlated with *ITGB3*, *VEGFR2*, *DLL4*, *YAP* and *WWTR1* in LUSC (*Figure 7D*), BLCA, and STAD (*Figure 7—figure supplement 1B,D*). And the expression of angiogenic factors including VEGFR2 and Tie2 is significantly correlated with the expression of *CCN1* in these cancer patient cohorts (*Figure 7—figure supplement 1E*). However, age showed no significant difference between *CCN1* high or *CCN1* low patient group in LUSC and STAD cohorts, whereas, in BLCA, age and stage of cancer showed significant (p<0.05) difference or enrichment between *CCN1* high and *CCN1* low patient groups, suggesting potential association of age and stage of cancer with patient survival in BLCA (*Figure 7—figure supplement 1F–H*). Taken together, these results suggest that overexpression of YAP/TAZ increases CCN1 expression and secretion from cancer cells into the

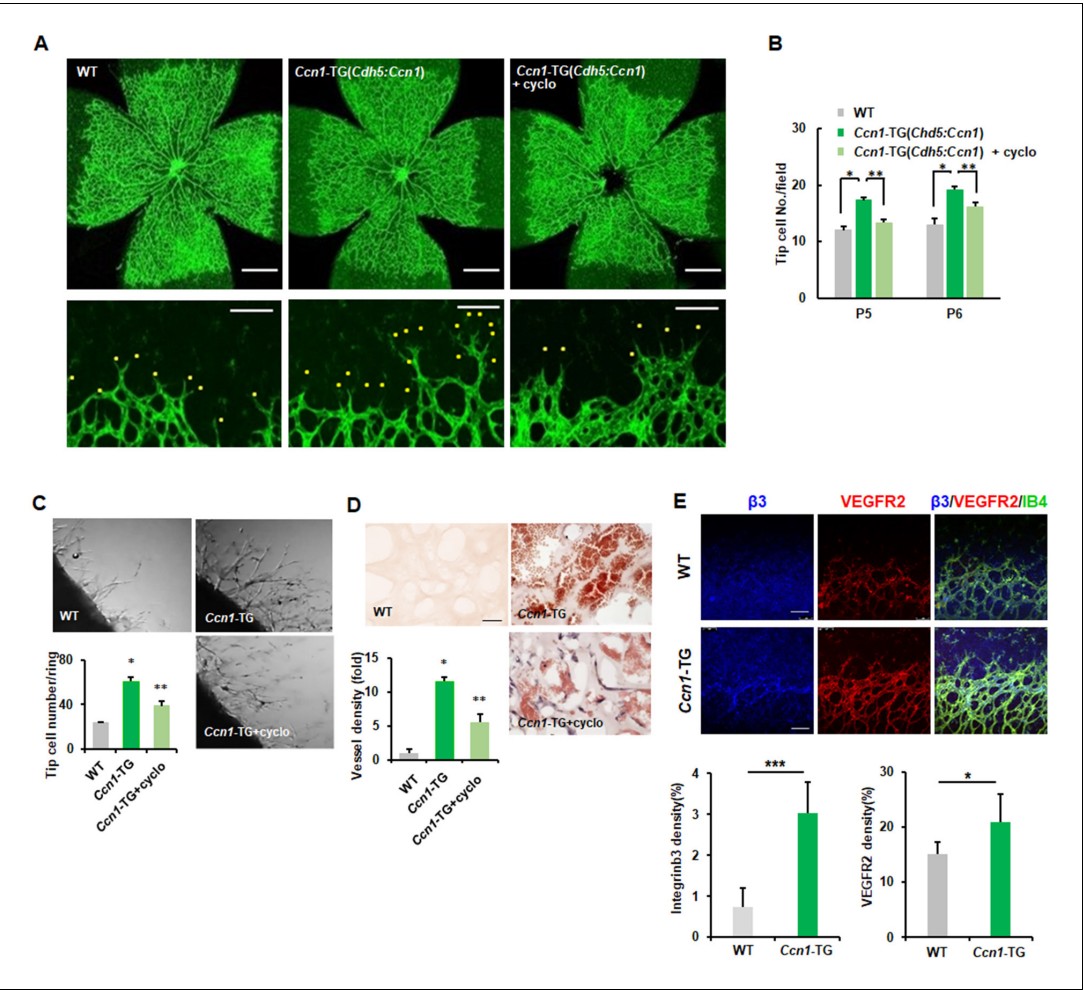

**Figure 6.** Endothelial CCN1 is crucial for sprouting angiogenesis. (**A, B**) Whole mount retinas at postnatal day 5 (P5) stained with anti-IB4 antibody, allowing comparison of whole retina (upper) and peripheral retina ECs (lower) in WT and TG mice. Yellow dots represent sprouting tip cells. Tip cell number was counted and shown in B. Scale bar = 50 μm (upper) and 100 μm (lower). *p<0.001, **p<0.01. (**C**) Mouse aortas from WT and *Ccn1*-TG mice were plated on Matrigel-coated plates and treated with cyclo(RGDfK) (1 μg/mL). Tip cells among sprouting ECs were counted and graphed. Three independent experiments were performed (n = 3/group). Magnification = 5 × . (**D**) Matrigel was inoculated into flanks of WT or *Ccn1*-TG mice and treated with cyclo(RGDfK) (20 mg/kg). After 7 days, Matrigel plugs were isolated, sectioned, and stained with H and E. Magnification = 400 × . Scale bar = 100 μm. Vessel density was quantified and graphed. *p<0.001 vs. Control, **p<0.01 vs. TG. (**E**) Expression of integrin β3 and VEGFR2 was determined via immunofluorescence and quantified in P5 retinas of WT and *Ccn1*-TG mice. Scale bar = 50 μm. ***p<0.001, *p<0.05 vs. Control.
DOI: https://doi.org/10.7554/eLife.46012.016

The following figure supplements are available for figure 6:

**Figure supplement 1.** Whole mount retinas at postnatal day 5 (p5) stained with anti-IB4 and CCN1 antibody in WT and TG mice.
DOI: https://doi.org/10.7554/eLife.46012.017
**Figure supplement 2.** CCN1 induces in vivo angiogenesis.
DOI: https://doi.org/10.7554/eLife.46012.018

tumour microenvironment, which activates tip cell activity by increasing the expression of YAP/TAZ as well as VEGFR2 and DLL4 in ECs to promote vascular sprouting and tumour angiogenesis. Our results indicate that CCN1 expression induced by activated YAP/TAZ increases tip cell activity in ECs

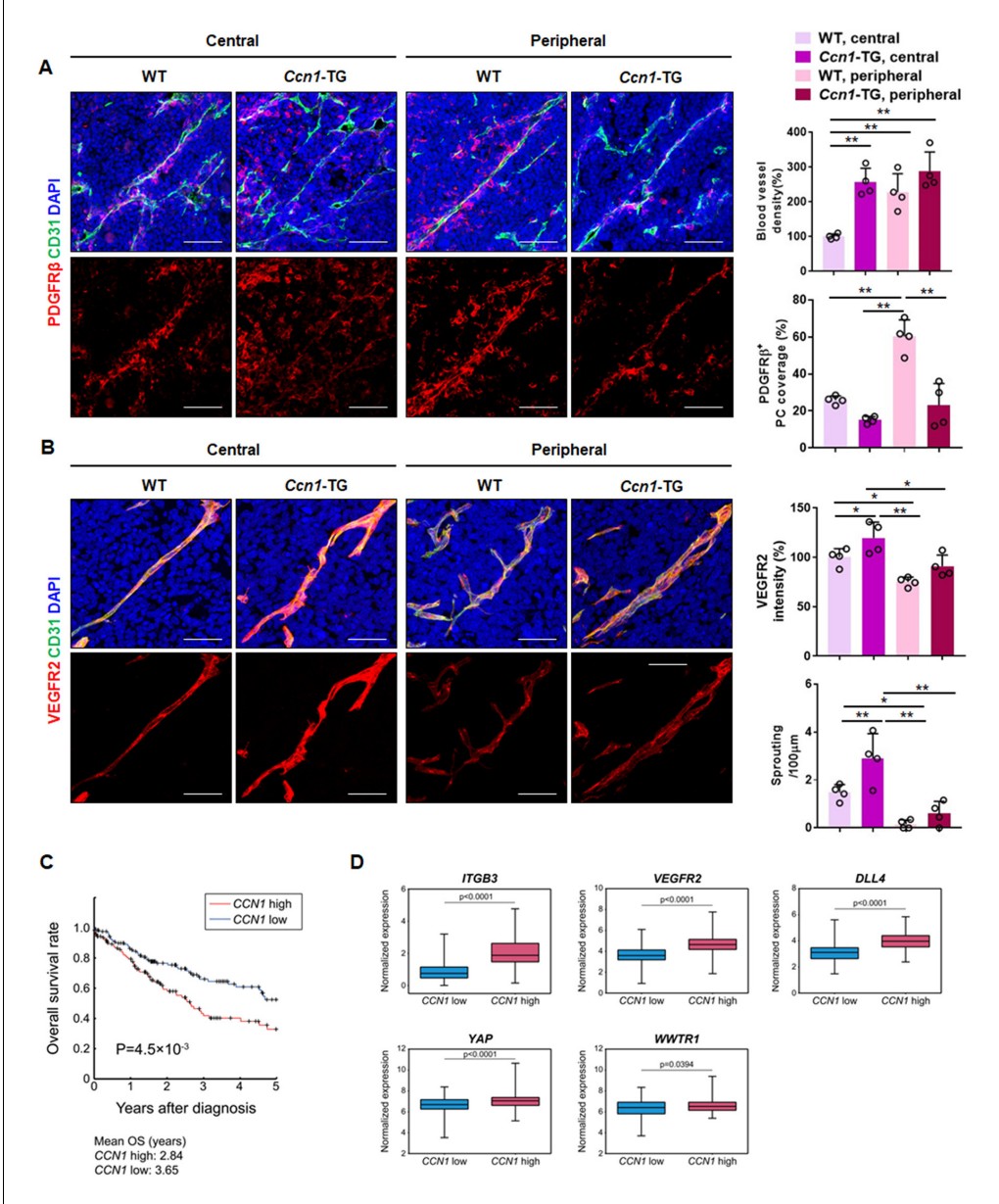

**Figure 7.** Role of CCN1 in pathological angiogenesis and clinical correlation with integrin/Hippo pathway. (**A**) LLC allograft tumour tissues in WT and *Ccn1*-EC-specific TG mice were extracted at 10 days after inoculation, sectioned, and immunostained with anti-CD31 and anti-PDGFβ antibodies and green and red fluorescently labelled secondary antibodies. Stained cells were quantified in the central and peripheral regions. Scale bar = 100 μm. *p<0.05, **p<0.01. (**B**) The same sectioned tissues as in (**A**) immunostained with anti-VEGFR2 and CD31. Quantitation was performed in the central and peripheral regions of the tissues. The number of sprouting sites was normalised to the total vessel length. Scale bar = 50 μm. *p<0.05, **p<0.01. (**C**) The difference in the survival curves of the two groups, *CCN1* high (red line) and *CCN1* low (blue line) 25% of patients with highest and lowest mRNA expression levels, respectively, n = 126/group] was evaluated using log-rank test with Kaplan–Meier estimation. p=8.7 $\times$ $10^{-3}$ between two groups. (**D**) Box plots of normalised mRNA expression levels of *ITGB3, VEGFR2, DLL4, YAP*, and *WWTR1* in *CCN1*-high and *CCN1*-low groups.

DOI: https://doi.org/10.7554/eLife.46012.019

The following figure supplement is available for figure 7:

**Figure supplement 1.** Negative correlation between mRNA expression levels of *CCN1* and patient survival in several cancer.

DOI: https://doi.org/10.7554/eLife.46012.020

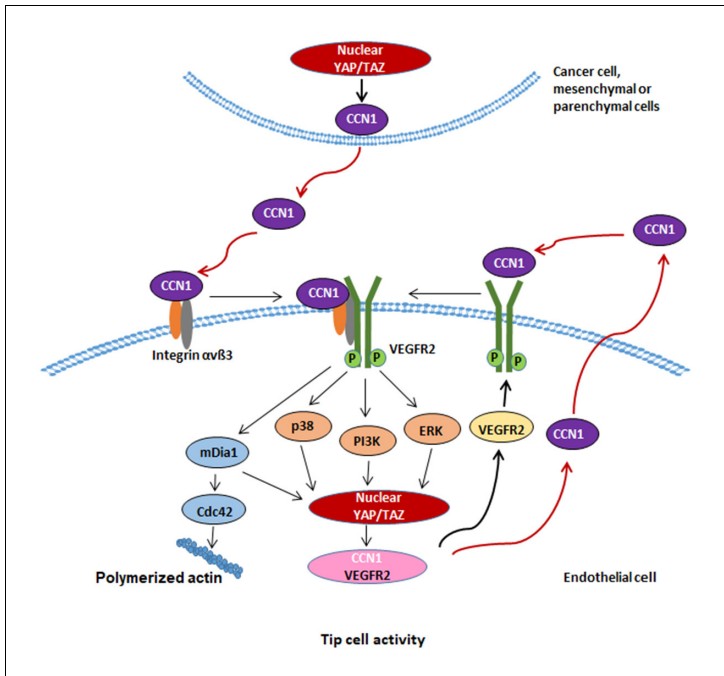

**Figure 8.** Schematic diagram depicting CCN1 induction of tip cell identity through interplay of integrin αvβ3/ VEGFR2 with Hippo pathway. CCN1 in the milieu is sensed by ECs by binding to membrane integrin αvβ3 and VEGFR2 receptors, thus activating the MAPK and PI3K-YAP/TAZ signalling pathways, leading to activation of YAP/ TAZ. YAP/TAZ transfer to the nucleus facilitates transcriptional regulation of CCN1, which may sustain the CCN1- VEGFR2-MAPK, PI3K-YAP/TAZ signalling circuit for maintaining tip cell activity in ECs. In addition, CCN1-induced mDia1 is necessary for YAP/TAZ activation and leads to induce filopodia formation through increased Cdc42 Rho GTPase activity.

DOI: https://doi.org/10.7554/eLife.46012.021

via filopodial formation mediated by the mDia1 Rho effector. CCN1 increases activation of the VEGFR2, ERK1/2, P38 MAPK, and PI3K signalling pathways by binding with integrin αvβ3 and VEGFR2. These signalling pathways subsequently induce YAP/TAZ activity and CCN1 expression, sustaining tip cell activity in ECs (*Figure 8*).

## Discussion

Secretion of CCN1 by tumours can facilitate tumour angiogenesis (*Maity et al., 2014*), and knockdown of *ccn1* in zebrafish resulted in defective vessels in terms of migration and development, especially sprouting behaviour. Based on these data, we attributed the angiogenic defects of CCN1 knockdown to the dysfunction of a specific type of EC: the tip cells. Tip cells migrate and branch according to increased concentrations of VEGF, whereas exposure to relatively low concentrations of VEGF induces the proliferation of stalk cells. Tip/stalk cell fate might be determined by different responses for cells to the environmental cue. It has been indicated that the main determinant for tip/ stalk cell fate is the Notch signalling pathway (*Hellström et al., 2007*; *Blanco and Gerhardt, 2013*; *Boareto et al., 2015*). Tip cell specification is initially achieved by DLL4 expression in the topmost cells and by Notch signalling in adjacent ECs. Tip cells that respond to VEGF signalling by increasing their levels of DLL4 expression have the strongest reactions to VEGF and acquire many filopodial phenotypes; in contrast, increased expression levels of Notch and Jagged-1 in neighbouring cells (stalk cells) suppress tip cell-like behaviour (*Carmeliet et al., 2009*). Increased expression of DLL4 is a key determinant for tip cell specification. DLL4/Notch1 signalling between adjacent ECs establishes an adequate ratio between tip cells and stalk cells in response to VEGF (*Hellström et al., 2007*). Here, we found that CCN1 increased tip cell phenotype and the expression of the tip cell

marker gene *DLL4*, possibly mediated through the activation of VEGFR2 by CCN1. A new tip cell marker CD34 expressing ECs show low proliferation (*Siemerink et al., 2012*), and CD34 has been expressed in undifferentiated cells with stem-cell properties, such as hematopoietic stem and progenitor cells (*Krause et al., 1996*; *Nielsen and McNagny, 2009*), suggesting that tip cells rarely divide but have stem-like properties among ECs. CD34 expressed in sprouting aorta and spheroid ECs and CD34+/DLL4+ expressing ECs were increased by CCN1 treatment, confirming the CCN1-induced tip cell formation.

CCN1 is highly induced by hypoxia and is regulated via both hypoxia-inducible factor 1α (HIF-1α)-dependent and -independent mechanisms (*Kunz et al., 2003*; *Meyuhas et al., 2008*; *Wolf et al., 2010*). Since hypoxia enhances VEGF expression—in addition to regulating ECM composition, deposition, posttranslational modifications, and rearrangements (*Germain et al., 2010*)—it is possible that hypoxic conditions increase the formation of tip cells, followed by stalk cell proliferation and tube formation, to supply oxygen to cells through neovessels. Furthermore, hypoxia-induced CCN1 and VEGF can collaboratively activate tip cell induction in ECs. If branched vessels are formed, oxygenation lowers the levels of CCN1 or VEGF, resulting in the survival of quiescent phalanx cells (*Carmeliet et al., 2009*). It has been reported that CCN1 regulates the expression of VEGF-A and VEGF-C in fibroblasts (*Chen et al., 2001*; *Mo et al., 2002*). We did not observe increased levels of VEGF, either in the sera of EC-specific *Ccn1*-TG mice or in EC culture media treated with CCN1, which is consistent with previous results (*Chintala et al., 2015*). This discrepancy may be due to the EC-specific targeted effects. Therefore, these results suggest that CCN1 induces tip cell activity in ECs in the absence of VEGF through VEGFR2 activation and in collaboration with integrin αvβ3 (*Figures 2*, *3* and *6*). Interestingly, during embryonic development, the expression patterns of VEGF and CCN1 almost overlap in the notochord, limbs, developing nervous system (including the ventral spinal cord, midbrain, and hair follicles), myocardium, truncus arteriosus, and dorsal aorta (*Dumont et al., 1995*; *Miquerol et al., 1999*; *Latinkic et al., 2001*; *Mo and Lau, 2006*; *Krupska et al., 2015*). The expression of these proteins in certain mesenchymal and vascular regions, such as the dorsal aorta and placental vessels, overlaps with FLK1 (VEGFR2) expression (*Dumont et al., 1995*), suggesting that CCN1 may function in collaboration with VEGF and/or VEGFR2, depending on the cellular and developmental context.

During development, CCN1 is also widely expressed in the cardiovascular system (*Kireeva et al., 1997*). Thus, CCN1-deficient mice provide some insight into the vital roles of CCN1 in cardiovascular development. The majority of *Ccn1*-null embryos die of haemorrhage and/or placental defects between E11.5 and E14.5 (*Mo et al., 2002*). Moreover, vascular defects in chorioallantoic fusion—by under-vascularisation in the placental labyrinth or large vessel bifurcation—have been found in *Ccn1*-null embryos, along with dilated large foetal vessels, such as the aorta (*Mo et al., 2002*). However, in P5 mice, the EC-specific deletion of CCN1 results in an increase in the number of tip cells in retinal ECs (*Chintala et al., 2015*). This result may reflect the high expression and secretion of CCN1 in other cell types, increasing tip cell activity in ECs. CCN1 is expressed in most cell types but not in ECs after development (*Su et al., 2004*). It is expressed in the front of the sprouting retina vessel, as previously reported (*Chintala et al., 2015*), and it functions in both paracrine and autocrine manner (*Latinkic et al., 2001*; *Mo and Lau, 2006*; *Krupska et al., 2015*). For the first time, after we confirmed that CCN1 is expressed only in tip cells in an autocrine manner, we revealed the mechanism by which tip cell activity is sustained. We treated cells with recombinant CCN1 primarily to mimic paracrine CCN1. In addition, we used EC-specific *Ccn1*-overexpressing mice to observe the function of endothelial CCN1. In recent research focusing on the role of some genes in blood vessels, mice with EC-specific gene knockout (KO) have generally been used; however, since the target gene in this case was more weakly expressed in ECs than in other surrounding cells, we cannot rule out the possibility that EC-specific KO phenotypes may be affected by proteins expressed in the other cell types, especially when studying a secreted protein, such as CCN1. Therefore, even though the results from EC-specific KO mice demonstrate dramatic effects on abnormal vascular formation, it is crucial to figure out the endogenous functions of genes on the basis of their expression patterns.

It has been reported that CCN1 does not induce the proliferation of fibroblasts or ECs (*Kireeva et al., 1996*), suggesting that CCN1 cannot enhance stalk cell phenotype, as seen in our experiments. Expression of *VEGFR2, DLL4* and *SOX17* was significantly induced by CCN1 or VEGF. These findings are in agreement with a recent report that SOX17/VEGFR2 constitutes a feedback loop activated by VEGF signalling (*Kim et al., 2016*). SOX17 is a transcription factor expressed in tip

cells that enhances developmental angiogenesis (*Lee et al., 2014*). Therefore, we speculate that CCN1 is involved in vascular sprouting by increasing the SOX17-VEGFR2 feedback loop, resulting in tip cell rather than stalk cell specification.

Direct physical interactions between purified CCN1 and integrin αvβ3 have already been observed in cell-free systems, and the EC adhesion induced by CCN1 is mediated by its binding to integrin αvβ3 (*Kireeva et al., 1998*). In fact, several members of the CCN protein familyinduce angiogenic processes, such as proliferation, chemotaxis, and tube formation in ECs, through binding to integrin αvβ3 (*Babic et al., 1998*; *Kubota and Takigawa, 2007*). In ECs, we found that CCN1 induced interactions between CCN1, integrin αvβ3, and VEGFR2. These interactions may then influence tip cell formation and tip cell activity induced by CCN1, indicating that CCN1 functions as a matricellular protein that mediates cell–matrix communication (*Yang and Lau, 1991*). In addition, reactivation of Hippo pathway in several cancers on inhibition of integrin linked kinase (*Serrano et al., 2013*) and repressed CCN1 on targeted inhibition of integrin αvβ3 suppressing the metastasis in osteosarcoma (*Gvozdenovic et al., 2016*) suggested interplay of integrin αvβ3 and Hippo signalling but this is the first time to show the CCN1 mediated feedback loop through integrin αvβ3/YAP/TAZ axis in vascular sprouting.

Here, we demonstrated that CCN1 plays a key role in the induction of tip cell identity in EC sprouting. The inhibition of DLL4 in tip cells induces a loss of Notch signalling and is associated with an increase in VEGFR2 and VEGFR3 in stalk cells (*Shawber et al., 2007*; *Suchting et al., 2007*). The inhibition of DLL4/Notch signalling causes large vessels and increases vascular density in tumours (*Li et al., 2011*). The loss of CCN1 in ECs appears to inhibit DLL4/Notch signalling, even in the presence of VEGF, in retinal vessel formation (*Chintala et al., 2015*). Therefore, we speculate that CCN1 may increase tip cell activity via the DLL4/Notch signalling pathway. Interestingly, CCN1 increases the expression of both DLL4 (a negative feedback regulator of VEGF) and SOX17 (a positive regulator of VEGF). Thus, CCN1 may be a key molecule modulating VEGF signalling, either in the presence or absence of VEGF. Therefore, it is likely that the involvement of CCN1 in the activation of VEGFR2 is a key link in VEGFR2 feedback regulation (*Kim et al., 2016*).

The increased phosphorylation of VEGFR2 (Y1175) caused by the loss of CCN1 in ECs (*Chintala et al., 2015*) may induce EC proliferation because of the dual roles of Y1175 phosphorylation in both EC proliferation and EC migration. Our results reflect the paracrine effects of CCN1 treatment on the Y1175 phosphorylation of VEGFR2 in EC migration, initiated by tip cells (*Figures 2*, *3* and *4*). Y1054 phosphorylation, involved in tyrosine kinase activity, clearly led to CCN1-mediated increases in VEGFR2 kinase activity levels, thus enhancing downstream signalling and leading to the expression of tip cell marker protein DLL4.

Accumulating evidence indicates the existence of many overlapping roles for CCN1 and VEGF in vascular formation (*Ferrara et al., 2003*; *Krupska et al., 2015*). Recently, it was reported that EC-specific targeting of CCN1 increased the proliferation of retina ECs, resulting in dense and enlarged vascular networks that lacked the normal arrangement of arterioles, capillaries, and venules (*Chintala et al., 2015*). Hyperplasia of ECs and an increased venule diameter can be explained by enhanced stalk cell activity. Additionally, increased numbers of filopodia and decreased DLL4 signalling activity have been observed in the frontal cells of the retina following EC-specific *Ccn1* KO, suggesting that loss of CCN1 in ECs disturbs the Notch/DLL4 signalling pathway (*Chintala et al., 2015*). Thus, CCN1 may distinctly regulate the activities of tip cells and stalk cells.

Disrupting angiogenic vessel formation, either by inhibition of VEGF itself or by inhibition of the VEGF signalling pathway, limits targeting efficiency due to increases in intratumoural hypoxia and the induction of other growth factors, such as basic fibroblast growth factor (bFGF), stromal-derived factor 1 (SDF-1), and Tie-2 (*Rapisarda and Melillo, 2012*). Furthermore, DLL4/Notch signalling has been found to mediate this resistance (*Li et al., 2011*). Novel and successful methods for targeting tumour angiogenesis have recently been developed. For example, combining anti-VEGF signalling therapy with an EGFR inhibitor (*Larsen et al., 2011*) or kinase inhibitors (*Moreno Garcia et al., 2012*) improves clinical outcomes. Thus, studies of specialised endothelial tip, stalk, and phalanx cell in developing sprouting vessels may provide approaches for eliminating resistance to anti-VEGF therapy (Avastin) and increasing the efficiency of treatments for angiogenic diseases.

# Materials and methods

## Key resources table

| Reagent type(species) or resource | Designation | Source or reference | Identifiers | Additional information |
|---|---|---|---|---|
| Antibody | anti-Flk1(VEGFR2) (A-3) (Mouse monoclonal) | Santa Cruz | sc6251 RRID:AB_628431 | WB(1:1000) |
| Antibody | anti-DLL4(H-70) (Rabbit polyclonal) | Santa Cruz | sc28915 RRID: AB_2092978 | WB(1:1000) |
| Antibody | anti-Sox17 (Goat polyclonal) | R and D Systems | AF1924 RRID:AB_355060 | WB(1:1000) |
| Antibody | anti-p-VEGFR2 (Tyr1059)(D5A6) (Rabbit monoclonal) | Cell Signalling | 3817 RRID:AB_2132351 | WB(1:1000) |
| Antibody | anti-p-VEGFR2 (Tyr1175) (19A10) (Rabbit monoclonal) | Cell Signalling | 2478 RRID:AB_331377 | WB(1:1000) |
| Antibody | anti-p-ERK1/2 (Thr202/Tyr204) (197G2) (Rabbit monoclonal) | Cell Signalling | 4377 RRID:AB_331775 | WB(1:1000) |
| CyeAntibody | anti-ERK1/2 (137F5) (Rabbit monoclonal) | Cell Signalling | 4695 RRID:AB_390779 | WB(1:1000) |
| Antibody | anti-p-p38 (Thr180/Tyr182) (Rabbit polyclonal) | Cell Signalling | 9211 RRID:AB_331641 | WB(1:1000) |
| Antibody | anti-p38 (A-12) (Mouse monoclonal) | Santa Cruz | sc7972 RRID:AB_620879 | WB(1:1000) |
| Antibody | anti- p85$\alpha$ (Z-8) (rabbit polyclonal) | Santa Cruz | sc423 RRID:AB_632211 | WB(1:1000) |
| Antibody | anti- p-PI3-kinase p85 $\alpha$ (Tyr 508) (goat polyclonal) | Santa Cruz | sc12929 RRID:AB_2252313 | WB(1:1000) |
| Antibody | anti-CCN1 (Rabbit polyclonal) | Abcam | Ab24448 RRID: AB_2088724 | WB(1:1000) IHC (1:200) |
| Antibody | anti-CCN1 (Rabbit polyclonal) | Novus Biologicals | NB100-356 RRID: AB_10000986 | (1:1000) neutralization |
| Antibody | anti-integrin $\alpha$v$\beta$3 (Mouse monoclonal) | Novus Biologicals | NB600-1342 RRID: AB_10003443 | WB(1:1000) |
| Antibody | anti-p-STAT-3 (Tyr705) (D3A7) (Rabbit monoclonal) | Cell signalling | 9145 RRID: AB_2491009 | WB(1:1000) |
| Antibody | anti-p-STAT1(Tyr701) (58D6) (Rabbit monoclonal) | Cell signalling | 9167 RRID: AB_561284 | WB(1:1000) |
| Antibody | anti-STAT3 (79D7) (Rabbit monoclonal) | Cell signalling | 4904 RRID: AB_331269 | WB(1:1000) |
| Antibody | anti-STAT1 (Rabbit polyclonal) | Cell signalling | 9172 RRID:AB_2198300 | WB(1:1000) |
| Antibody | anti-p-YAP (Ser127) (Rabbit polyclonal) | Cell signalling | 4911 RRID: AB_2218913 | WB(1:1000) |
| Antibody | anti-YAP/TAZ (D24E4) (Rabbit monoclonal) | Cell signalling | 8418 RRID: AB_10950494 | WB(1:1000) |

*Continued on next page*

*Continued*

| Reagent type(species) or resource | Designation | Source or reference | Identifiers | Additional information |
|---|---|---|---|---|
| Antibody | anti-p-LATS1 (Thr1079)(D57D3) (Rabbit monoclonal) | Cell signalling | 8654 RRID: AB_10971635 | WB(1:1000) IF (1:200) |
| Antibody | anti-LATS1(C66B5) (Rabbit monoclonal) | Cell signalling | 3477 RRID:AB_2133513 | WB(1:1000) |
| Antibody | Alexa-488–conjugated anti-isolectin B4 | Invitrogen | I21411 RRID:AB_2314662 | IHC (1:200) |
| Antibody | anti-VEGF (Goat polyclonal) | R and D Systems | AF564 RRID:AB_2212821 | (1:1000) neutralization |
| Antibody | anti-vinculin (EPR8185) (Rabbit monoclonal) | Abcam | ab196579 RRID:AB_2810877 | IF (1:200) |
| Antibody | Cdc42 (Mouse monoclonal) | Cell signalling | 8747 RRID:AB_2810881 | WB(1:1000) |
| Antibody | Integrin β3 blocking IgG (Mouse monoclonal) | Millipore | MAB1976 RRID:AB_2296419 | Neutralization |
| Antibody | Anti-GFP (B-2) (Mouse monoclonal) | Santa Cruz | sc9996 RRID:AB_627695 | WB(1:1000) |
| Antibody | anti-CD31 (Mouse monoclonal) | BD Biosciences | 553370 RRID:AB_2638986 | IF (1:200) |
| Antibody | Anti-CD34 (Rabbit polyclonal) | Boster | PA1334 RRID:AB_2810878 | IF (1:200) |
| Antibody | Anti-DLL4 (Rat-monoclonal) | R and D systems | MAB1389 RRID:AB_2092985 | IF (1:200) |
| Other | Alexa-488-phalloidin | Invitrogen | A12379 | IHC (1:200) |
| Chemical compound, drug | JNK Inhibitor II anthra[1,9 cd]pyrazol-6(2H)-one-1,9-pyrazoloanthrone (SP600125) | Calbiochem | 420119 | |
| Chemical compound, drug | P38 MAP Kinase Inhibitor 4-(4-fluorophenyl)—2-(4-methylsulfinylphenyl)—5-(4-pyridyl)1H-imidazole (SB203580) | Calbiochem | 559389 | |
| Chemical compound, drug | 2'-amino-3'-methoxyflavone (PD98059) | Calbiochem | 513000 | |
| Chemical compound, drug | 2-(4-morpholinyl)—8-phenyl-4H-1-benzopyran-4-one (LY294002) | Calbiochem | 440202 | |
| Chemical compound, drug | Verteporfin | Sigma-Aldrich | SML0534 | |
| Chemical compound, drug | Cyclo(RGDfK) | Selleckchem | S7834 | |
| Peptide recombinant protein | CCN1 | R and D Systems | 4055 | |

*Continued on next page*

*Continued*

| Reagent type(species) or resource | Designation | Source or reference | Identifiers | Additional information |
|---|---|---|---|---|
| Peptide recombinant protein | Vitronectin | Corning | 354238 | |
| Peptide recombinant protein | fibronectin | Corning | 354008 | |
| Commercial assay, or kit | Active Cdc42 detection kit | Cell Signalling | 11859 | |
| Recombinant DNA reagent | pEG-DIAPH1 | Pro Jung Weon Lee (College of Pharmacy, Seoul National University) | | |
| Recombinant DNA reagent | pEG-DIAPH1 ΔN3 | Prof. Jung Weon Lee (College of Pharmacy, Seoul National University) | | |
| Recombinant DNA reagent | pGL3-CCN1 (cloned on VE-cadherin promoter) | This paper | | Progenitors: PCR, pGL3 |
| Recombinant DNA reagent | pGL3-DLL4 luciferase plasmid | Prof. Young Geun Kwon Yonsei University | | |
| Recombinant DNA reagent | pRL-SV40 plasmid | Promega | E2231 | |
| Sequence-based reagent | siRNA:siCCN1 | Qiagen | SI03053477 SI03028655 SI02626428 SI02626421 | |
| Sequence-based reagent | siRNA: siIntegrin β3 | Qiagen | SI02664095 SI02628094 SI00034188 SI00034174 | |
| Sequence-based reagent | Negative control siRNA | Qiagen | SI03650325 | |
| Cell line (*H. sapiens*) | HUVEC | ATCC | CRL-1730 RRID:CVCL_2959 | |
| Cell line (*H. sapiens*) | EA.hy926 | ATCC | CRL-2922 RRID:CVCL_3901 | |
| Cell line (*M. musculus*) | LCC | ATCC | CRL-1642 RRID:CVCL_4358 | |
| Transgenic mice | *Ccn1*-TG (Chd5:Ccn1, C57BL/6J) | Macrogen, Seoul, Korea | | |
| Transgenic zebrafish (*D. rerio*) | hemizygous TG (*flk*-1:*EGFP*)[s843] | Korea Zebrafish Organogenesis Mutant Bank (ZOMB) at Kyungpook National University *Jin et al., 2005* | | |
| Sequence-based reagent | ccn1 MO (*D. rerio*) 5′-CTCCGCTGACACAC ACACACAGGAC-3′ | Gene-Tools, Philomath | | ccn1-l2, ENSDARG00000099985 |
| Software, algorithm | Prism | Graphpad Software | RRID: SCR_002798 | |
| Software, algorithm | ImageJ | https://imagej. nih.gov/ij/ | RRID: SCR_003070 | |

*Continued on next page*

*Continued*

| Reagent type(species) or resource | Designation | Source or reference | Identifiers | Additional information |
|---|---|---|---|---|
| Software, algorithm | edgeR package | http://www.bioconductor.org./help/search/index.html?q=edger+package/ **Robinson et al., 2010** | RRID: SCR_012802 | |
| Software, algorithm | quantile normalisation | http://bioconductor.org **Bolstad et al., 2003** | RRID: SCR_001786 | |
| Software, algorithm | Kaplan–Meier estimation | https://www.xlstat.com/en/solutions/features/kaplan-meier-analysis **Bewick et al., 2004** | | |
| NCI Genomic Data Commons (GDC) | Data Portal | https://gdc-portal.nci.nih.gov **Grossman et al., 2016** | | |

## Zebrafish strains and MO-mediated gene knockdown in zebrafish

Wild-type AB strain and Tg(*flk-1:EGFP*)[s843] zebrafish were used in this study (*Jin et al., 2005*). Adult Zebrafish and embryos were raised at 28°C on a 14 hr light and 10 hr dark cycle. The two kinds of *ccn1* MO was synthesised by Gene-Tools (Philomath, OR, USA) with the following sequence: ccn1l2 splicing MO: 5'-CTCCGCTGACACACACACACAGGAC-3', ccn1l2 ATG MO: 5'-TCCTCTGATTTC TTCCAGAATGCAT-3' (ccn1-l2, ENSDARG00000099985). For control MO, prepared control oligo provided from Gene-Tools was used with the following sequence: Standard MO: 5'- CCTCTTACC TCAGTTACAATTTATA-3'. All of the MOs was suspended in Nuclease-Free deionized water and injected into 1 cell stage TG embryos. Blood vessels were observed under a Nikon A1 plus confocal microscope or under a ZEISS Axioskop confocal microscope (Oberkochen, Germany). For sense RNA synthesis, *ccn1* cDNA was ligated into pCS2 vector, linearized by Not I restriction enzyme and transcribed by Sp6 RNA polymerase provided from Invitrogen mMESSAGE kit. *ccn1* sense RNA was injected into 1 cell stage TG embryos with ccn1l2 splicing MO.

## Whole mount in situ hybridization in zebrafish

Embryos were fixed with 4% paraformaldehyde (PFA) in phosphate-buffered saline (PBS) overnight, rinsed with PBST (0.1% Triton X-100 in PBS), transferred to methanol, and stored at −20°C until use. After rehydration with PBST, embryos were incubated with proteinase K solution. Embryos were refixed with 4% PFA for 20 min and then washed with PBST four times. Embryos were then pre-hybridised and hybridised at 65–67°C with antisense RNA probes. Subsequently, non-binding anti-sense RNA probes were washed, and embryos were transferred to blocking solution (5% sheep serum) for 2 hr at room temperature (RT). An alkaline phosphatase-conjugated sheep anti-digoxigenin Fab' fragment (Roche, Basel, Switzerland) was added (1:4000 dilution) and incubated for 3 hr. Embryos were then washed five times with PBST for 15 min each at RT and then 2 times for 15 min each with staining buffer. Next, BCIP/NBT was added, and embryos were incubated at RT until staining.

## Cell culture

HUVECs (ATCC, Manassas, VA, USA) were cultured on gelatine-coated (1% gelatine, Sigma-Aldrich) tissue culture plates in EBM2 (Lonza, Basel, Switzerland) supplemented with EGM2 kit and 20% foetal bovine serum (FBS, Hyclone) in a humidified atmosphere of 5% $CO_2$ at 37°C. Human endothelial EA. hy926 cells (ATCC) were maintained in DMEM with 10% FBS and 1 × antibiotics.

## Immunofluorescent assays

Aorta, spheroids or cells after treatment with or without CCN1 were fixed with 4% paraformaldehyde (PFA) and incubated overnight with primary antibodies. The next day, they were washed several times and further incubated with secondary antibodies with specific fluorescent probes and visualised using a confocal microscope (Leica TCS SP5 II Dichroic/CS, Germany).

## siRNA-mediated gene knockdown in ECs

A pool of four distinct siRNAs directed against *CCN1* and *ITGB3* was purchased from Qiagen (Valencia, CA, USA). HUVECs were transfected with *CCN1* and *ITGB3* siRNA using a microporator (Invitrogen). Gene silencing was examined by performing semiquantitative RT-PCR at 24 hr after transfection.

## Matrix adhesion assays

HUVECs were suspended in serum-free M199 medium and allowed to adhere to vitronectin (5 μg/mL, Corning)- or fibronectin (1 μg/mL, Corning)-coated and 0.1% BSA/PBS-blocked 96-well plates in the presence or absence of integrin β3 blocking IgG (1 μg/mL, Millipore) for 90 min. After washing with PBS, adhered ECs were stained with Crystal Violet for 10 min and lysed with 1% SDS, and the absorbance (O.D.) was measured at 550 nm.

## Tube formation assay

HUVECs were seeded in a 48-well Matrigel (10 mg/mL, BD Biosciences, San Diego, CA, USA)-coated culture plate for 30 min at 37°C. Tube formation was observed for 12 hr, and the numbers of branching points in the tubular structures were counted under a microscope and photographed at 100 × magnification. Two or three independent experiments were performed and statistically analysed.

## Aortic ring assay

An aortic arch obtained from 6 week-old C57BL/6J WT or *Ccn1*-TG mice was cut into ~1.0 mm pieces. The aortic rings were placed in 48-well Matrigel or collagen-coated plates. Then, 100 μL of EBM-2 medium containing VEGF or CCN1 was added to each well. The formation of vascular sprouts was observed after 7 days. The ECs expressing with DLL4, CD34 and CD31 were determined as tip cells by immunofluorescent assay and counted under the confocal microscope (Leica TCS SP5 II Dichroic/CS, Germany).

## Spheroid sprouting assay

HUVECs ($4 \times 10^3$) were mixed with a 1.2% methylcellulose solution (Sigma-Aldrich) and M199 medium containing 0.1% FBS. The cell suspension was distributed into a 96-well U-shaped culture plate (BD Biosciences) and cultured overnight. The next day, the spheroids were harvested and mixed with diluted rat tail collagen (BD Biosciences). One millilitre of the spheroid-containing collagen solution was aliquoted into each well of a pre-warmed 24-well culture plate. After 1 hr for polymerization of the collagen at 37°C, VEGF or CCN1 was applied to the polymerized collagen gel, and endothelial sprouting was observed and fixed with 4% PFA for IF after 48 hr.

## Wound migration assay

Confluent EA.hy926 cells in 12-well plates (Corning) were starved in 1% FBS in DMEM for 12 hr and treated with mitomycin C (0.5 μg/mL) for 1 hr. A wound was made with a pipette tip, and the cell debris was removed by washing with serum-free medium. The cells were treated with VEGF (10 ng/mL), CCN1 (10 ng/mL), or DMSO vehicle. The number of cells that moved beyond the injury line was counted. Three independent experiments were performed.

## Matrigel plug assay

Matrigel (300 μL, BD Biosciences) was inoculated subcutaneously into both flanks of WT or EC-specific *Ccn1*-TG (*Chd5:Ccn1*) mice treated with vehicle or cyclo(RGDfK) (20 mg/kg). Matrigel plugs were removed at 7 days after inoculation and fixed with 4% paraformaldehyde (PFA, pH 7.4). Serial sections (8 μm) were stained with H and E and observed under a light microscope (Zeiss). The vessel density in each section was quantified after taking pictures, and the erythrocyte-containing vessel area was measured based on image analysis using ImageJ (National Institutes of Health, Bethesda, MD, USA).

## Western blotting analysis

Proteins were separated by SDS-PAGE and transferred to a nitrocellulose membrane (Whatman, Maidstone, UK). The membrane was blocked with 5% non-fat milk in Tris-buffered saline (TBS) containing 0.1% Tween-20 for 30 min at RT. The membrane was incubated with primary antibody at 4°C overnight and then with horseradish peroxidase-conjugated secondary antibodies for 1 hr. The membrane was visualised using West Pico Chemiluminescent Substrate (PIERCE, Woburn, MA, USA).

## Semiquantitative and real-time RT-PCR

Total RNA was extracted from cultured cells using TRIzol Reagent (Invitrogen). cDNA synthesis from total RNA was performed using a first-strand cDNA synthesis kit for reverse transcription-PCR according to the manufacturer's protocol (TOYOBO, Osaka, Japan). The cDNA was used as a template for PCR using specific primers for *CCN1* (forward, 5'-GAGCACATGTTACTGCTTCA-3'; reverse, 5'-GATAGCTGCCTCTCACAGAC-3') and *zEF-1a* (forward, 5'-GCAAGGAGAAGACCCACATC-3'; reverse, 5'-CTTGAACCTCGGCATGTTG-3'). The conditions for semiquantitative PCR were 30 cycles of denaturation (94°C for 30 s), annealing (50°C for 40 s), and extension (72°C for 40 s), followed by a final extension (72°C for 10 min). Real-time PCR was performed using a Bio-Rad Real-Time PCR system with SYBR Green PCR Master Mix (Applied Biosystems, Foster City, CA, USA). The specific primer sequences for real-time PCR are shown in *Table 1*.

## In situ PLA

To confirm the presence of protein–protein interactions, HUVECs were fixed with 4% PFA for 15 min and PLA was performed following the manufacturer's instructions (Sigma-Aldrich). Briefly, chamber slides were blocked; incubated with anti-VEGFR2, anti-CCN1, and anti-integrin β3 antibodies; and then incubated with PLA probes overnight at 4°C. After washing, cells were incubated with the respective secondary antibodies conjugated with the PLA probe for 1 hr at 37°C, washed, and ligated for 30 min at 37°C. Subsequently, amplification with polymerase was performed for 100 min at 37°C. To detect the nuclei of the HUVECs, 4'−6-diamidino-2-phenylindole (DAPI) was used for staining for 1 min at RT, and the slides were mounted using Vectashield (Vector Laboratories, Burlingame, CA, USA). Red fluorescence signals were visualised with a confocal laser scanning microscope (LSM 700, Zeiss). The number of red dots in each cell was quantified using ImageJ.

## Dual luciferase reporter assay

HUVECs were transfected with *pGL3-DLL4* luciferase and *Renilla* luciferase pRL-SV40 plasmids by electroporation (Thermo Fisher Scientific, Waltham, MA, USA). One day after transfection, cells were treated with VEGF (10 ng/mL) or CCN1 (10 ng/mL) in the presence or absence of cyclo(RGDfK) (5 μM) for 24 hr. Luciferase assays were performed using a Dual-Glo Luciferase Reporter Assay System (Promega).

**Table 1.** Gene primer sequences for qPCR.

| Gene name | Forward | Reverse |
| --- | --- | --- |
| *DLL4* | ATTCGTCACCTGGATCCTTC | TCATTCTGGGCCAGTTGTAA |
| *SOX17* | CAGACTCCTGGGTTTTTGTTGTTGCTG | GAAATGGAGGAAGCTGTTTTGGGACAC |
| *ROBO4* | GACGGGAATCAGAACCACTT | CAGAGAAACACAGGCCAAGA |
| *VEGFR2* | CTCGGGTCCATTTCAAATCT | GCTGTCCCAGGAAATTCTGT |
| *JAG1* | AAGGCTTCACGGGAACATAC | AGCCGTCACTACAGATGCAC |
| *NOTCH1* | AAGATGCTCCAGCAACACAG | GGCTCTGGCAAGTCTCCTAC |
| *β-actin* | GGATGTCCACGTCACACTTC | CACTCTTCCAGCCTTCCTTC |
| *DIAPH1* | CAAGACAACCTCTTGTGCCC | GCTCCGAAGCTAGCAGAGAT |

DOI: https://doi.org/10.7554/eLife.46012.022

## Detection of active Cdc42

HUVECs were seeded at a density of $2 \times 10^5$ in a 100 mm dish. The next day, cells were transfected with empty vector, *DIAPH1* WT, or *DIAPH1* double negative mutant for 24 hr and further starved overnight. Then, cells were treated with PBS or CCN1 (10 ng/mL) for 30 min. Protein lysate was collected by centrifugation at $16,000 \times g$ at 4C for 15 min. Active Cdc42 was extracted from the lysate per the manufacturer's protocol. Briefly, an equal amount of protein was used for GTPase assay (Active Cdc42 detection kit, 8819, Cell Signalling Technology). GST-PAK1-PBD fusion protein was used to bind the activated form of GTP-bound Cdc42. The level of activated Cdc42 was checked by WB with Cdc42 antibody (Cell Signalling Technology).

## Generation of *Ccn1*-TG mice

The *Cdh5* (VE-cadherin) promoter fragment was cloned according to a strategy previously described (*Rönicke et al., 1996*; *Kappel et al., 1999*) in a pGL3 basic vector (Promega, Madison, WI, USA). Briefly, after using PCR for synthesis with an additional restriction site, *EcoRV* following *SmaI*, *Cdh5* promoter fragments were inserted between the *SacI* and *SmaI* sites. The mouse cDNA was synthesised from total RNA, and the *Ccn1* fragments were amplified by PCR with primers including the restriction sites *EcoRV/XbaI* at each end and inserted into a *Cdh5* promoter-cloned pGL3 vector. TG mice were generated by microinjection of the *Cdh5* promoter-driven *Ccn1* expression plasmid into fertilised C57BL/6 mouse oocytes. Following microinjection, the oocytes were transferred to the oviducts of pseudo-pregnant C57BL/6 females (Macrogen, Seoul, Korea). To identify TG mice, genomic DNA was extracted from the mouse tails and amplified using PCR with primers targeting the middle of the *Cdh5* promoter to the middle of the *Ccn1* gene. C57BL/6 mice (Japan SLC, Inc, Hamamatsu, Japan) were maintained under specific pathogen-free conditions with a constant humidity and temperature at 26 °C, filtered air and a 12/12 hr light/dark cycle. Mice were treated with $CO_2$ by inhalation in a chamber for anaesthesia, immediately prior to sacrifice.

## In vivo tumour allograft experiment and IHC

LLC cells ($2 \times 10^5$) were injected subcutaneously into the right and left flanks of 6-week-old WT and *Ccn1*-TG (*Chd5:Ccn1*) C57BL/6J mice. When the tumour mass reached around 500 mm³, we excised the mass and fixed it with 4% PFA. After tissues were processed for frozen sample preparation, samples were embedded in OCT compound and frozen quickly by liquid nitrogen. Frozen blocks were cut into 50 µm sections, blocked with 5% goat serum in PBST (0.03% Triton X-100 in PBS), and then incubated for 3 hr at RT with primary antibodies.

## Patient survival analysis and mRNA expression correlation

To evaluate the correlation between mRNA expression levels of *CCN1* and patient survival in several cancers including LUSC, we first collected read counts for 60,483 gene features for 502 primary tumour samples of LUSC, 375 samples of STAD, and 414 samples of BLCA from the NCI Genomic Data Commons (GDC) Data Portal (*Grossman et al., 2016*). The read counts were normalised using the TMM normalisation method in the edgeR package (*Robinson et al., 2010*); the normalised counts were converted to $\log_2$ read-counts after adding one to the normalised counts, and the $\log_2$ read-counts were further normalised using the quantile normalisation method (*Bolstad et al., 2003*). Finally, we divided the samples into two groups, encompassing the top and bottom 25% of patients with the highest and lowest mRNA expression levels based on the normalised $\log_2$ read-counts and evaluated differences in survival between the two groups using the log-rank test with Kaplan–Meier estimation (*Bewick et al., 2004*).

## IHC for retinal angiogenesis analysis

Retinas were incubated with or without anti-CCN1 antibody (Abcam) overnight. After washing several times, the samples were incubated for four hat RT with Alexa Fluor488-conjugated isolectin B4 (IB4, Invitrogen) and anti-rabbit Alexa Fluor 647-conjugated IgG. The whole-mount retinas were visualised and digital images obtained using a Leica TCS SP5 II Dichroic/CS confocal microscope (Leica, Wetzlar, Germany). The number of front tip cells with filopodia was counted in three different peripheral fields of the retina. The radial length of the blood vessels in the postnatal retina was measured as the shortest distance from the optic nerve head to the peripheral vascular front in each

retinal quadrant. The vascular density in the whole-mounted retina was calculated as the IB4$^+$ microvessel area divided by the total measured area of the retina and presented as a percentage.

## Statistical analysis

ANOVA was used to assess the significance of intergroup differences. For the quantification of the IHC data, one-way analysis of variance with Newman-Keuls multiple comparison test. A $p$-value$<0.05$ was considered statistically significant and noted in each Figure legend, and results are represented as means ± standard deviations (SD). Transcript data from human cancer database (from GDC) was analyzed with t-test after edgeR package was employed for genomic data analysis. Overall survival rate was evaluated using log-rank test with Kaplan–Meier estimation. The correlation of mRNA expression levels between nine angiogenesis marker genes and *CCN1* in three cancers (LUSC, BLCA, and STAD) was estimated by Spearman's rank correlation (Spearman's rho) and empirical testing using the following procedure. For each cancer, we first randomly selected two genes among the total expressed genes and computed Spearman's rho between two genes. This random selection was repeated 100,000 times, resulting in 100,000 random Spearman's rho values. We then estimated an empirical null distribution for the Spearman's rho by applying the Gaussian kernel density estimation method (*Bowman and Azzalini, 1997*) to the 100,000 random Spearman's rho values. Finally, the p-value for the observed Spearman's rho was computed by the right-sided test using the empirical distribution.

# Additional information

## Funding

| Funder | Grant reference number | Author |
|---|---|---|
| National Research Foundation of Korea | NRF-2013R1A2A2A01068868 | You Mie Lee |
| National Research Foundation of Korea | NRF-2017R1A2B3002227 | You Mie Lee |

The funders had no role in study design, data collection and interpretation, or the decision to submit the work for publication.

## Author contributions

Myo-Hyeon Park, Ae kyung Kim, Gun-Hyuk Jang, Do Young Hyeon, Data curation, Formal analysis, Writing—review and editing; Sarala Manandhar, Data curation, Formal analysis, Writing—original draft, Writing—review and editing; Su-Young Oh, Data curation, Formal analysis; Li Kang, Dong-Won Lee, Sun-Hee Lee, Hye Eun Lee, Sang Heon Suh, Data curation; Tae-Lin Huh, Hae-Chul Park, Methodology, Project administration; Daehee Hwang, Data curation, Formal analysis, Project administration; Kyunghee Byun, Data curation, Formal analysis, Project administration, Writing—review and editing; You Mie Lee, Conceptualization, Formal analysis, Funding acquisition, Methodology, Writing—original draft, Writing—review and editing

## Author ORCIDs

You Mie Lee https://orcid.org/0000-0002-5756-7169

## Ethics

Animal experimentation: The animal handling and experimental procedures were strictly conducted as per the Guidelines for Care and Use of Laboratory Animals issued by the Institutional Ethical Animal Care Committee of Kyungpook National University.

## Decision letter and Author response

Decision letter https://doi.org/10.7554/eLife.46012.025
Author response https://doi.org/10.7554/eLife.46012.026

## Additional files

**Supplementary files**
• Transparent reporting form
DOI: https://doi.org/10.7554/eLife.46012.023

**Data availability**

All data generated or analysed during this study are included in the manuscript and supporting files.

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
