## [Decision Letter]

Thank you for submitting your article "Cyr61 interlinks integrin and Hippo pathway to autoregulate tip cell activity" for consideration by *eLife*. Your article has been reviewed by three peer reviewers, and the evaluation has been overseen by a Reviewing Editor and Jonathan Cooper as the Senior Editor. The following individual involved in review of your submission has agreed to reveal their identity: Hong Chen (Reviewer #2).

It would be important for authors to either show a CRISPR/Cas9 phenotype or to perform a rescue experiment – that is to use a MO that targets the endogenous gene but not a rescue construct, and put both in and show a rescue. A more detailed quantitative analysis of the overall state of the embryos subjected to morpholinos relative to controls would also strengthen this work.

Major points:

1) The conclusion of the present study on the role of Cyr61 in angiogenesis is inconsistent with a previous study (Chintala et al., 2015). It is possible that the vascular phenotypes could differ, given that the present study and the previous study have employed different approaches (gain-of-function versus loss-of-function). However, it is still necessary to reconcile this disparate results/conclusion with more convincing evidences. For example, examining the vascular phenotypes in the EC-specific Cyr61-deleted mouse would be a definitive approach.

2) The reviewers raised numerous issues and errors in imaging presentation, descriptions, and statistical analyses. The authors should carefully go through their manuscript and correct them. Moreover, there were several technical issues that could be solved with better and advanced approaches.

*Reviewer #1:*

While the presented study is interesting, it is mainly based in vitro observation. The in vivo experiments performed in Zebrafish and mouse would require further work and characterization to completely support the proposed model by the authors.

In particular, in the last sentence of the Results section the authors write: Our results indicate that Cyr61 expression induced by activated YAP/TAZ increases tip cell activity in ECs via filopodial formation mediated by the mDi1 Roh effector." Cyr61 increases activation of VEGFR2, ERK1/, P38 MAPK and PI3K signaling pathways by binding with integrin alphavbeta3 and VEGFR2. These signalling pathways subsequently induce YAP/TAZ activity and Cyr61 expression, sustaining tip cell activity in ECs (Figure 8)". Even if Cyr61 is not highly expressed in retina tip cells, the authors should generate the EC-specific knockout mice in order to clearly confirm their proposed model.

*Reviewer #2:*

The investigators of this manuscript reported that Cyr61 positively regulated endothelial tip cell activation and promoted angiogenesis through integrin αvβ3 and YAP/TAZ pathway and its potential clinical implications on the survival of cancer patients. Thus, the area of research holds substantial translational value and clinical relevance. The idea to study the role of integrin αvβ3 and YAP/TAZ pathway in mediating the effect of Cyr61 on angiogenesis is original. The experimental design was logical with the usage of Zebrafish MO and of endothelial cell-specific Cyr61 transgenic mice. A variety of studies were used to confirm endothelial tip cell function and angiogenesis in vivo and in vitro. The investigators also analyzed human patient data from NCI Genomic Data Commons (GDC) Data Portal with lung squamous cell carcinoma, bladder urothelial cancer, and stomach adenocarcinoma to establish the reverse relationship between patient survival and Cyr61 expression. These data were strong and supportive to the conclusion in general. However, the in-depth mechanisms underlying how Cyr61 modulates activity of integrin αvβ3 and signaling capacity of VEGFR2 is less clear. Further, Zebrafish MO studies need to be complemented with rescue experiments or knockout approaches.

Additionally, there are a few concerns in the manuscript that need to be addressed.

1) Very little data were provided on the patients including age, sex, stage of cancer, other critical medical conditions like DM, HTN, coronary artery disease that may have a substantial impact on the survival. The labeling in Figure 7C and Figure 7—figure supplement 1A and C (Days to death) appeared to be wrong; the mean survival age needs to be indicated. Information of basal time point for the survival curve needs to be explained whether "0" indicates the date of diagnosis, surgery, or treatment.

2) The authors stated that Cyr61 was only expressed in tip cells in an autocrine manner, not in stalk cells. It was also not produced in endothelial cells after development. It would be nice to include discussion regards the differences between tip cells and stalk cells and the potential mechanisms regulating its expression.

3) WB results lack statistical analyses and some blots do not reflect the results. It would be helpful to replace with better representative and unsaturated gel bands throughout the manuscript.

4) Image data appeared to be adjusted artificially and inappropriately. For example, integrin β3 signal in the merged image does not reflect or display information shown in the corresponding single channel (Figure 5—figure supplement 1C).

5) The description for the statistical method is unclear. The statistical analysis section needs to be addressed more appropriately. For instance, "Analysis of variance" means "ANOVA" where it indicated as two different approaches. IHC data was indicated where one-way ANOVA with post-hoc test was used. Authors need to explain how other experiments were analyzed. Some experiments contain two groups where One-way ANOVA may not be suitable for statistical analysis. Figure 2C and Figure 5E have two groups and should be analyzed by t-test. Transcript data from human cancer database (from GDC) should be analyzed with t-test if edgeR or DESeq was employed for genomic data analysis.

6) Many places showed the unaligned or different thickness of error bars within the graphs, which need to be fixed (Figure 2A, 2B, 2C, 2E, 3B, 5B, G, H, I). Also, there are missing error bars in some place (Supplemental figure B).

7) VEGFR2 was used as an internal loading control after VEGF or Cyr61 stimulation. VEGFR2 is not a proper loading control as VEGFR2 phosphorylation can induce VEGFR2 internalization and degradation.

8) Since α-SMA is expressed in smooth muscle cells and myofibroblasts. α-SMA alone is not a solid pericyte marker (Figure 7A). Therefore, the conclusion that Cyr61TG decrease pericyte coverage is overstating. PDGFR-β or NG2 should be used for validating pericyte coverage in transplanted tumor section.

9) Materials and methods and resource table showed that BAECs were used in this study (subsection “Cell culture” and key resources table). However, BAECs had not been used throughout the study. It would be nice to state which experiments BAECs were used.

10) In Figure 1 legend, it should be "intron 1/exon2" as indicated in text and schematic figure, not "intron2/exon3 boundary".

*Reviewer #3:*

This paper by Kim et al. investigates the angiogenic function of cyr61 (CCN1) using numerous in vivo and in vitro models. They conclude that cyr61 induces tip cells by working thru VEGFR and integrin signaling, and they place YAP/TAZ in the signaling axis. Although there are some potentially interesting insights provided, there are significant concerns with both novelty, significance, and methodology that significantly diminish enthusiasm.

1) The overall novelty is not considered high, and the work is predicted to have moderate to minimal impact on the field. It is known that Cyr61 is involved in angiogenesis, and a 2015 Development paper (cited) describes an angiogenic loss-of-function phenotype suggesting that Cyr61 negatively regulates VEGFR2 and integrin signaling with some underlying mechanism. This work's unique aspect of YAP/TAZ involvement is predicted from recent work of others showing that YAP/TAZ regulates angiogenesis.

2) The premise for the experiments is not always clear. EC cell-autonomy vs. non cell-autonomy for Cyr61 function is often interchanged, so it is not clear how the authors interpret their data. Most experiments are gain of function and show that exogenous protein or EC-specific expression increase sprouting parameters, yet Chintala et al. showed that vascular-specific deletion of CCN1 also lead to increased sprouting, and this apparent discrepancy is not explained. The authors claim little to no endothelial expression of cyr61, but Chintala et al. showed expression at the angiogenic front. The single in vivo loss-of-function experiment uses only one Zfish morpholine, and the phenotype is pleiotropic without a rescue, so in my opinion cannot be interpreted. Thus the premise for the work is not strong. The authors assume that "tip cell" is a fate, not a phenotype, and use primarily 2D EC cultures to look at perturbations they associate with "tip cells".

3) There are numerous methodological issues that weaken the work. For example, the Zfish experiment cannot be interpreted as the embryos are severely compromised and there is no rescue done; the aortic ring assay does not use a marker to distinguish endothelial cells from other cells that also sprout; the spheroid assay is not robust and does not form luminized vessels in controls; the assay for filopodia is primarily done in 2D, where it is not clear what is measured or how filopodia are normalized.

---

## [Author Response]

Major points:1) The conclusion of the present study on the role of Cyr61 in angiogenesis is inconsistent with a previous study (Chintala et al., 2015). It is possible that the vascular phenotypes could differ, given that the present study and the previous study have employed different approaches (gain-of-function versus loss-of-function). However, it is still necessary to reconcile this disparate results/conclusion with more convincing evidences. For example, examining the vascular phenotypes in the EC-specific Cyr61-deleted mouse would be a definitive approach.

Thank you for your valuable comment. As for these concerns, first, many previous publications showed and suggested that CCN1 (Cyr61) enhances angiogenesis (reviewed in (Jun and Lau, 2011)), consistent with our results, but inconsistent with another study (Chintalaet al., 2015). CCN1 stimulates cell migration or chemotaxis in fibroblasts and endothelial cells, and promote the invasiveness of certain cancer cells (Chen and Lau, 2009). CCN1, CCN2 and CCN3 are functioning through direct binding to integrin αvβ3 in endothelial cells to promote proliferation and induce chemotaxis and formation of tubules (Babicet al., 1998; Kubota and Takigawa, 2007; Jun and Lau, 2011). The previous publications found that these CCNs may be involved in embryonic development, inflammatory diseases and tumorigenesis through their angiogenic activities (Kubota and Takigawa, 2007; Kularet al., 2011).

In addition, Ccn1 enhances tumor growth through its potent angiogenic activity (Babicet al., 1998), accordingly, forced expression of Ccn1 in breast cancer cells promotes tumor growth in xenografts with increased vascularization (Xieet al., 2001; Tsaiet al., 2002). The results are correlated with our results for LLC allograft mouse tumor model (Figure 7A-B) and TCGA big data analysis (Figure 7D).

Second, a question struck in our mind, if *CCN1* is rarely expressed in ECs after birth but expressed in other CNS cells and fibroblast (Kireevaet al., 1997), in *Ccn1*-EC-specific KO mouse, other cells can secrete CCN1 into ECs which can bind to integrin αvβ3 leading to the increased angiogenesis. Hence, the increased angiogenesis in *Ccn1*-EC-specific KO mouse is not likely owing to the inhibitory function of CCN1 towards angiogenesis via inhibition of VEGFR2 activity in ECs (Chintala et al., 2015), but is likely to the angiogenic function of the secreted CCN1 from non-EC cells where it might have been expressed. Thus, EC-specific KO for certain protein which is rarely expressed in EC but fairly expressed in non-ECs cannot demonstrate the role of that protein in EC if it is secreted, which we have discussed in “Discussion” part in our manuscript.

Chintala et al., 2015, also showed that Ccn1 is expressed only in the front area of the retina, but scarcely expressed in other retinal vascular ECs at p5. This expression pattern in retina may support our results for the autonomous activation of tip cells through the YAP/TAZ (which are expressed in retinal vessels at P5 (Kimet al., 2017)) activity induced by Ccn1 (Figure 3 and 5). Also EC-specific KO of Lats (an upstream inhibitor or YAP/TAZ) in mice showed the significant increase in “*Ccn1* expression signature” in ECs (Kimet al., 2017), suggesting that YAP/TAZ activation increases Ccn1 in the front region of retinal vessels.

In conclusion, we would like to suggest that our approaches of in vitro study and in vivo EC-specific transgenic mouse are towards the right direction to identify the exact role of CCN1 in angiogenic process. Therefore, lots of evidences from our in vitro and in vivo research with CCN1 on ECs as well as the role of YAP/TAZ, upstream regulator of CCN1, in sprouting angiogenesis, indicate that CCN1 is the angiogenic inducer as a tip cell activator but not the angiogenic inhibitor.

Third, unfortunately, currently we do not have the mouse system for inducible EC-specific deletion of *Ccn1*, however, we have the zebrafish system for showing the role of ccn1 in endothelial cells. Also, due to the reasons mentioned above, we adopted two different strategies as below to confirm our results, rather than an EC-specific KO mice adopted by Chintala et al., 2015.

Zebrafish knockdown system with another type of morpholino against *ccn1* (targeting ATG transcription starting site) in vascular specific GFP-expressing embryos was performed. We obtained the vascular defects same as the results from previous morpholino targeting splicing region (between the intron 1/exon 2 boundary) (Figure 1C).

We confirmed these results with a rescue experiment by microinjecting *ccn1* cRNA and obtained the results with normal vascular development (Figure 1C).

2) The reviewers raised numerous issues and errors in imaging presentation, descriptions, and statistical analyses. The authors should carefully go through their manuscript and correct them. Moreover, there were several technical issues that could be solved with better and advanced approaches.

Thank you very much for kind directions for the errors. It was the initial submission recommended by *eLife* for initial review process. We solved the issues and problems raised by reviewers by responding each reviewer’s comments.

Reviewer #1:

*While the presented study is interesting, it is mainly based* in vitro *observation. The in vivo experiments performed in Zebrafish and mouse would require further work and characterization to completely support the proposed model by the authors.*

In particular, in the last sentence of the Results section the authors write: Our results indicate that Cyr61 expression induced by activated YAP/TAZ increases tip cell activity in ECs via filopodial formation mediated by the mDi1 Roh effector." Cyr61 increases activation of VEGFR2, ERK1/, P38 MAPK and PI3K signaling pathways by binding with integrin alphavbeta3 and VEGFR2. These signalling pathways subsequently induce YAP/TAZ activity and Cyr61 expression, sustaining tip cell activity in ECs (Figure 8)". Even if Cyr61 is not highly expressed in retina tip cells, the authors should generate the EC-specific knockout mice in order to clearly confirm their proposed model.

Thank you for your valuable comment. We were allowed to do the zebrafish morpholino (MO) using additional MO and rescue experiments within 3 months by Associate Editor, as we do not have the mouse system for inducible EC-specific deletion of *Ccn1* unfortunately, but, we have the zebrafish system for showing the role of ccn1 in endothelial cells. Due to the reasons mentioned above, we adopted two different strategies as below to confirm our results, rather than an EC-specific KO mice.

Zebrafish knockdown system with another type of morpholino against *ccn1* (targeting ATG transcription starting site) in vascular specific GFP-expressing embryos was adopted. We obtained the vascular defects same as the results from previous morpholino targeting splicing region (between the intron 1/exon 2 boundary) and added the results in Figure 1C (V-VI) and Figure 1D.

We confirmed these results with a rescue experiment by microinjecting *ccn1* cRNA and got normal vascular development results and images (Figure 1C, VII-VIII) and Figure 1D.

As a result, we found the sprouting defects of vasculature both in ATG morpholino (MO) (Figure 1C, V-VI) and in splicing region of intron1/exon2 site (Figure 1C, III-IV), respectively. And these sprouting defects were significantly rescued by the sense mRNAof *ccn1* (Figure 1C, VII-VIII). These results showed that ccn1 is responsible for the sprouting of vessel formation.

Reviewer #2:

*The investigators of this manuscript reported that Cyr61 positively regulated endothelial tip cell activation and promoted angiogenesis through integrin αvβ3 and YAP/TAZ pathway and its potential clinical implications on the survival of cancer patients. Thus, the area of research holds substantial translational value and clinical relevance. The idea to study the role of integrin αvβ3 and YAP/TAZ pathway in mediating the effect of Cyr61 on angiogenesis is original. The experimental design was logical with the usage of Zebrafish MO and of endothelial cell-specific Cyr61 transgenic mice. A variety of studies were used to confirm endothelial tip cell function and angiogenesis* in vivo *and* in vitro*. The investigators also analyzed human patient data from NCI Genomic Data Commons (GDC) Data Portal with lung squamous cell carcinoma, bladder urothelial cancer, and stomach adenocarcinoma to establish the reverse relationship between patient survival and Cyr61 expression. These data were strong and supportive to the conclusion in general. However, the in-depth mechanisms underlying how Cyr61 modulates activity of integrin αvβ3 and signaling capacity of VEGFR2 is less clear. Further, Zebrafish MO studies need to be complemented with rescue experiments or knockout approaches.*

In this paper, we checked only for the involvement of integrin αβ heterodimer not for the integrin α subunit specifically, because the previous publications suggest that integrin αvβ heterodimers play role in CCN1-induced proangiogenic activity. In a previous report, CCN1 has 20 AA residues (NCKHQCTCIDGAVGCIPLCP) which are binding sites for integrin αvβ3 for its proangiogenic activity (Chenet al., 2004). As mentioned in the Results section, β1 integrin is known to be involved in the tubular formation of migration of ECs by CCN1, but involvement of β1 was not found in the adhesion induced by CCN1 (Figure 5—figure supplement 1). We, thus, investigated the interaction between integrin αvβ3 and CCN1, VEGFR2 and integrin αvβ3 as well as CCN1. We then used specific peptide inhibitor against integrin αvβ3, cyclo (RGDfK), which binds specifically and with high affinity to αvβ3 to inhibit its activity (Lucieet al., 2009; Limet al., 2018). Cyclo (RGDfK) treated ECs abolished CCN1-treated VEGFR2 signaling pathway and Yap/Taz activation. Therefore, we concluded that VEGFR2 signaling pathways are activated by CCN1 via binding of integrin αvβ3/VEGFR2/CCN1. However, in-depth mechanism underlying how CCN1 modulates integrin ɑ and VEGFR2 activity is to be identified in separate further studies.

Additionally, there are a few concerns in the manuscript that need to be addressed.1) Very little data were provided on the patients including age, sex, stage of cancer, other critical medical conditions like DM, HTN, coronary artery disease that may have a substantial impact on the survival. The labeling in Figure 7C and Figure 7—figure supplement 1A and C (Days to death) appeared to be wrong; the mean survival age needs to be indicated. Information of basal time point for the survival curve needs to be explained whether "0" indicates the date of diagnosis, surgery, or treatment.

As suggested, we obtained clinical information of the patients from the original papers. Also, we assessed whether clinical parameters (age, sex, and stage of cancer) could have substantial impacts on the survival by performing 1) Student’s t-test to compare ages between *CCN1* high and *CCN1* low patient groups and 2) proportion tests for proportions of sex or stage of cancer in *CCN1* high and *CCN1* low patient groups. Age showed no significant difference between *CCN1* high or *CCN1* low patient group in LUSC and STAD cohorts. Also, none of sex and stage of cancer was significantly enriched in either *CCN1* high or *CCN1* low patient group in LUSC and STAD cohorts. Given that *CCN1* high and *CCN1* low patient groups showed significant difference in the survival, these data suggest that these parameters appear to have no significant impacts on patient survival. Interestingly, however, in BLCA, age and stage of cancer showed significant (P<0.05) difference or enrichment between *CCN1* high and *CCN1* low patient groups, suggesting potential association of age and stage of cancer with patient survival in BLCA. On the other hand, we could find other medical conditions including DM, HTN, and coronary artery disease from the original papers (TCGA data portal and supplementary tables) and thus could not assess potential impacts of these conditions on patient survival through the aforementioned proportional tests for their enrichments in either *CCN1* high or *CCN1* low patient group in LUSC, STAD, and BLCA cohorts. In the revised manuscript, we added the descriptions for these results in Results section (Figure 7—figure supplement 1).

Moreover, as suggested, we changed the x-axis label “Days to death (year)” in Figure 7C and Figure 7—figure supplement 1C to “Years after diagnosis”. We also included the mean survival age (mean OS; estimated as the area under the survival curve) of each patient group. Basal time point (“0”) for the survival curve indicates the date of the initial pathological diagnosis. We added this information to figure legends.

2) The authors stated that Cyr61 was only expressed in tip cells in an autocrine manner, not in stalk cells. It was also not produced in endothelial cells after development. It would be nice to include discussion regards the differences between tip cells and stalk cells and the potential mechanisms regulating its expression.

We discussed as much as possible about the differences between tip cells and stalk cells and the potential mechanisms regulating its specification by CCN1 in the Discussion section.

3) WB results lack statistical analyses and some blots do not reflect the results. It would be helpful to replace with better representative and unsaturated gel bands throughout the manuscript.

We performed Western blot (WB) analysis in Figure 3B again, quantified with previous blot data and graphed as shown in Figure 3B right panel. Other WB data were quantified and denoted as a Fold change value below each western band because of the space limitation.

4) Image data appeared to be adjusted artificially and inappropriately. For example, integrin β3 signal in the merged image does not reflect or display information shown in the corresponding single channel (Figure 5—figure supplement 1C).

Merged images are adjusted darker than the integrin β3 signals. We corrected it.

5) The description for the statistical method is unclear. The statistical analysis section needs to be addressed more appropriately. For instance, "Analysis of variance" means "ANOVA" where it indicated as two different approaches. IHC data was indicated where one-way ANOVA with post-hoc test was used. Authors need to explain how other experiments were analyzed. Some experiments contain two groups where One-way ANOVA may not be suitable for statistical analysis. Figure 2C and Figure 5E have two groups and should be analyzed by t-test. Transcript data from human cancer database (from GDC) should be analyzed with t-test if edgeR or DESeq was employed for genomic data analysis.

We described in detail the statistical analysis for the data with different approaches at the Materials and methods section. We used t-test for Figure 7D for the transcript analysis as reviewer indicated and added this in Materials and methods section as a “Statistical analysis” as well as in a part of figure legend.

6) Many places showed the unaligned or different thickness of error bars within the graphs, which need to be fixed (Figure 2A, 2B, 2C, 2E, 3B, 5B. G, H, I). Also, there are missing error bars in some place (Supplemental figure B).

We corrected unaligned graphs as well as graph format in all Figures.

7) VEGFR2 was used as an internal loading control after VEGF or Cyr61 stimulation. VEGFR2 is not a proper loading control as VEGFR2 phosphorylation can induce VEGFR2 internalization and degradation.

We used β-actin for a loading control in all Western blot analysis.

8) Since α-SMA is expressed in smooth muscle cells and myofibroblasts. α-SMA alone is not a solid pericyte marker (Figure 7A). Therefore, the conclusion that Cyr61TG decrease pericyte coverage is overstating. PDGFR-β or NG2 should be used for validating pericyte coverage in transplanted tumor section.

We used the PDFGR-β instead of α-SMA for validating pericyte coverage in transplanted tumor section.

9) Materials and methods and resource table showed that BAECs were used in this study (subsection “Cell culture” and key resources table). However, BAECs had not been used throughout the study. It would be nice to state which experiments BAECs were used.

After we used BAECs in the preliminary experiments, we changed the cells into HUVECs to do in vitro tube formation assay and obtained results shown in this manuscript. Therefore, we omitted this cells from the table.

10) In Figure 1 legend, it should be "intron 1/exon2" as indicated in text and schematic figure, not "intron2/exon3 boundary".

We corrected into intron 1/exon2.

Reviewer #3:This paper by Kim et al. investigates the angiogenic function of cyr61 (CCN1) using numerous in vivo and in vitro models. They conclude that cyr61 induces tip cells by working thru VEGFR and integrin signaling, and they place YAP/TAZ in the signaling axis. Although there are some potentially interesting insights provided, there are significant concerns with both novelty, significance, and methodology that significantly diminish enthusiasm.1) The overall novelty is not considered high, and the work is predicted to have moderate to minimal impact on the field. It is known that Cyr61 is involved in angiogenesis, and a 2015 Development paper (cited) describes an angiogenic loss-of-function phenotype suggesting that Cyr61 negatively regulates VEGFR2 and integrin signaling with some underlying mechanism. This work's unique aspect of YAP/TAZ involvement is predicted from recent work of others showing that YAP/TAZ regulates angiogenesis.

Thank you for your valuable comment. We responded these issues as described in the “Major points” from the Reviewing Editor above. Here we underscored the positive feedback regulation of tip cells by CCN1 and identified the signalling pathway from the membrane to transcription factor, YAP/TAZ in the nucleus. Formation or specification of tip cell should be continued among the ECs during active angiogenesis. Like stem cells, tip cells having migrating property may have to make stalk cells that follow them by dividing. Or only stalk cells are dividing behind the tip cells. Because the dynamic changes of Notch signalling as well as resultant genetic modification in migrating and extending cells, we still do not know exactly whether and how a cell at the topmost is maintained as a tip cell. We adopted the in vitro cell culture system where an angiogenic factor was applied to the whole culture media with the same concentration. Genetic changes or behaviour of ECs by the same manner even the migrating phenotypes were different. Therefore, we determined the expression level of tip cell markers as well as YAP/TAZ activity as a master regulator of tip cell. A receptor of CCN1, integrin αvβ3 made it associated into VEGFR2 to transfer signalling for activation of YAP/TAZ.

*2) The premise for the experiments is not always clear. EC cell-autonomy vs. non cell-autonomy for Cyr61 function is often interchanged, so it is not clear how the authors interpret their data. Most experiments are gain of function and show that exogenous protein or EC-specific expression increase sprouting parameters, yet Chintala et al. showed that vascular-specific deletion of CCN1 also lead to increased sprouting, and this apparent discrepancy is not explained. The authors claim little to no endothelial expression of cyr61, but Chintala et al. showed expression at the angiogenic front. The single* in vivo *loss-of-function experiment uses only one Zfish morpholine, and the phenotype is pleiotropic without a rescue, so in my opinion cannot be interpreted. Thus the premise for the work is not strong. The authors assume that "tip cell" is a fate, not a phenotype, and use primarily 2D EC cultures to look at perturbations they associate with "tip cells".*

Thank you for your valuable comment. We responded and performed the rescue experiment as described in “Response” to the “Major points” from the Associate Editor above. We tried to clarify the premise for the experiment in the whole manuscript as solid as possible, especially the EC-cell autonomy for CCN1 function.

At the beginning of research, in zebrafish knockdown experiments, we found that ccn1 has a role in the migration and sprouting of ECs during vascular development. Then using in vitro EC culture system, we identified the genetic changes of markers for specific ECs as well as the migrating ability after CCN1 was treated. Next, we investigated CCN1 induced YAP/TAZ and VEGFR2 activity, key regulators of vascular sprouting, and further signaling circuit to induce CCN1 again. Therefore, we emphasize that CCN1 may activate YAP/TAZ to induce target gene, CCN1 again for maintaining the tip cell properties. It was a gradual outcome, and our final conclusion was the auto-regulation by CCN1 in tip cells.

And we described and emphasized that CCN1 is expressed only in retina frontal EC, but not much in whole EC in the second paragraph of the Introduction section. And then we raised a question in the Results section about the discrepancy that CCN1 deletion increases sprouting even if it is expressed only in the frontal ECs in the subsection “CCN1 promotes endothelial sprouting activity in angiogenesis”..

3) There are numerous methodological issues that weaken the work. For example, the Zfish experiment cannot be interpreted as the embryos are severely compromised and there is no rescue done; the aortic ring assay does not use a marker to distinguish endothelial cells from other cells that also sprout; the spheroid assay is not robust and does not form luminized vessels in controls; the assay for filopodia is primarily done in 2D, where it is not clear what is measured or how filopodia are normalized.

Thank you for your valuable comment. We responded these issues as described in the “Major points” from the Reviewing Editor above. For the sprouting assay, we perform it again with 3D culture system.